# Ultrafast photoinduced C-H bond formation from two small inorganic molecules

Zhejun Jiang ®[1], Hao Huang[1], Chenxu Lu[1], Lianrong Zhou[1], Shengzhe Pan ®[1], Junjie Qiang[1], Menghang Shi[1], Zhengjun Ye[1], Peifen Lu[1], Hongcheng Ni ®[1,2] ✉, Wenbin Zhang ®[1] ✉ & Jian Wu ®[1,2,3,4] ✉

The formation of carbon-hydrogen (C-H) bonds via the reaction of small inorganic molecules is of great significance for understanding the fundamental transition from inorganic to organic matter, and thus the origin of life. Yet, the detailed mechanism of the C-H bond formation, particularly the time scale and molecular-level control of the dynamics, remain elusive. Here, we investigate the light-induced bimolecular reaction starting from a van der Waals molecular dimer composed of two small inorganic molecules, $H_2$ and CO. Employing reaction microscopy driven by a tailored two-color light field, we identify the pathways leading to C-H photobonding thereby producing $HCO^+$ ions, and achieve coherent control over the reaction dynamics. Using a femtosecond pump-probe scheme, we capture the ultrafast formation time, i.e., $198 \pm 16$ femtoseconds. The real-time visualization and coherent control of the dynamics contribute to a deeper understanding of the most fundamental bimolecular reactions responsible for C-H bond formation, thus contributing to elucidate the emergence of organic components in the universe.

The carbon–hydrogen (C-H) bond serves as a primary characteristic distinguishing inorganic and organic compounds in nature. As one of the most prevalent chemical bonds in organic compounds, the C-H bond serves as a vital link within the organic world and has been recognized as the building block for life. Understanding the cleavage, formation, and activation of C-H bonds is crucial in various fields, including organic synthesis[1,2], industrial chemistry[3,4], biochemistry[5–7], and interstellar chemistry[8,9]. The investigation of C-H bond formation from the reaction of small inorganic substances holds particular significance as it provides profound insights into the fundamental transition from inorganic to organic matter and establishes connections between interstellar chemical networks and the emergence of life[10,11]. Realizing coherent control of such C-H bond making process represents a significant advancement in the field of chemical organic synthesis.

Over the years, a plethora of studies have been devoted to inspecting the spectroscopic properties[9,12,13] and formation reactions[14–16] of C-H bonds, primarily relying on the reactive scattering measurements[17,18] and crossed-beam ion-neutral reactions[15,19]. Reactive collisions involving substances containing H and C atoms have been shown to produce radicals or compounds with C-H bonds[20,21]. Such reactions thus play a vital role in influencing the organic composition of the interstellar medium[22]. Scattering studies have enabled the extraction of reaction cross sections, kinetic energies, and angle-resolved state distributions of products containing C-H bonds through integrated beam measurements[18,23]. However, the detailed mechanism and dynamics of the C-H bond formation in bimolecular reactions, including the timescales and molecular-level coherent control, have long remained elusive, particularly in cases involving two small inorganic molecules.

[1]State Key Laboratory of Precision Spectroscopy, East China Normal University, Shanghai 200241, China. [2]Collaborative Innovation Center of Extreme Optics, Shanxi University, Taiyuan, Shanxi 030006, China. [3]Chongqing Key Laboratory of Precision Optics, Chongqing Institute of East China Normal University, Chongqing 401121, China. [4]CAS Center for Excellence in Ultra-intense Laser Science, Shanghai 201800, China. ✉e-mail: hcni@lps.ecnu.edu.cn; wbzhang@lps.ecnu.edu.cn; jwu@phy.ecnu.edu.cn

$H_2$ and CO molecules represent two prevalent small inorganic substances containing H and C atoms, respectively, and are widespread in nature. Due to their abundance in the interstellar environment, the reaction between $H_2$ and CO is considered a fundamental chemical process in the interstellar medium[24]. The bimolecular interaction of the neutral/ionized molecular system $H_2$ + CO can yield organic compounds with C–H bonds, such as the formyl ($HCO^+$) cation, which holds great importance in various fields, including atmospheric chemistry[25], combustion science[26], and astrochemistry[27]. Previous studies on such reactions have predominantly relied on full intermolecular collisions and scattering measurements[14,15,19], which, however, lack a definition of the spatiotemporal starting point for time-resolved studies. Experimental challenges associated with determining reaction time zero and the initial internuclear distance between the two reacting molecules have thus significantly impeded the investigation of the detailed mechanism and dynamics of bimolecular reaction processes. Moreover, the incoherent population of scattering states in collision measurements presents a significant obstacle in achieving coherent control of bond cleavage and formation in bimolecular reactions. To realize the coherent control, the precise manipulation of quantum interference of phase and amplitude of the wavefunctions of the reacting molecules is required[28–32].

Here, we investigate the formation of C–H bonds in light-induced bimolecular reactions of two inorganic molecules within a van der Waals (vdW) molecular dimer of $H_2$··CO. By analyzing the laser-phase-dependent directional emission of ions in a tailored two-color laser field, we identify the reaction channels responsible for C–H bond formation thereby producing $HCO^+$ and demonstrate the potential for all-optical coherent control of the bimolecular reaction. Additionally,

by performing femtosecond pump-probe measurements, we distinguish between a fast and a slow reaction pathway leading to C–H bond formation on the femtosecond timescale. The fast pathway, accessed via a two-site double ionization of the dimer, can be triggered by a single pulse. In contrast, the slow pathway involves an ion-neutral interaction of $H_2 + CO^+$ and takes about 198 fs. To gain further insights into the time-resolved dynamics of the reaction pathways, molecular dynamics simulations are conducted, with good agreements with experimental findings.

## Results

### Experimental approach

Compared to the beam scattering technique, the experimental approach of initiating a light-induced bimolecular reaction starting from a $H_2$··CO dimer offers unique advantages. Unlike bimolecular reactions in the beam scattering experiments, where the intermolecular distance is uncertain, the two interacting molecules within a molecular dimer system has a well-defined equilibrium geometry. Consequently, as pioneered by Witting's[33] and Zewail's[34] works, the starting point of the light-driven bimolecular reaction dynamics in a vdW dimer is well-defined and can be precisely tracked[35,36]. By employing ultrashort laser pulses with well-defined waveforms, it becomes possible not only to unambiguously time the dynamics but also to coherently control the reactions. In the experiment, the molecular beam containing the $H_2$··CO dimer was produced by supersonically expanding a gas mixture of neutral $H_2$ (95%) and CO (5%) molecules through a precooled nozzle. As depicted in Fig. 1a, the experimental measurements were performed using a cold target recoil ion momentum spectroscopy (COLTRIMS) setup[37,38], where the charged ions resulting from the bimolecular reaction within a $H_2$··CO

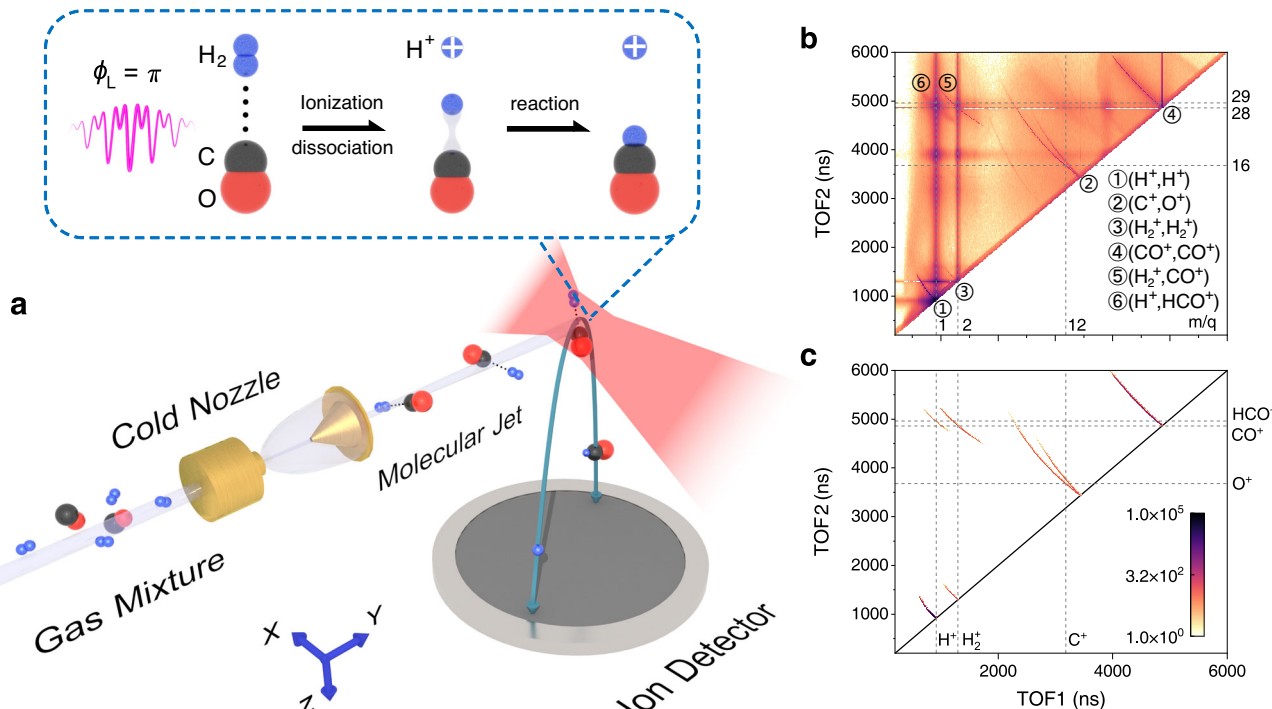

**Fig. 1 | Sketch of the experimental setup and time-of-flight coincidence map. a** Schematic illustration of the experimental configuration for investigating the light-induced bimolecular reaction of the $H_2$··CO dimer. The inset depicts the formation of the ($H^+$, $HCO^+$) ion pair from the bimolecular reaction of $H_2$ and CO within the dimer induced by a two-color laser field with a relative phase $\phi_L = \pi$. Upon interaction with the two-color pulse, the $H_2$··CO dimer is two-site doubly ionized, followed by the dissociation of $H_2^+$ into a $H^+$ (blue-plus sphere) and H atom (blue sphere). The asymmetric two-color pulse controls the reaction dynamics by driving

the emission of the H atom towards ($H^+$ away from) the C side of $CO^+$ (C atom: black sphere, O atom: red sphere), forming the $HCO^+$ ion with C–H bond. The red shadow depicts the laser focus interaction area. The cyan curve illustrates the trajectory of the ion fragments flying towards the detector. **b, c** Measured photoion-photoion coincidence spectrum. The horizontal and vertical axes correspond to the time-of-flight (TOF) of the first and second ions resulting from the two-body Coulomb explosion channels, respectively.

dimer are measured in coincidence. More details of the experiments can be found in the Methods section.

## Identifying and controlling the C−H bond formation channel

The identification of laser-induced bimolecular reaction channels can be achieved by examining the coincidence map of ionic products resulting from the reaction and subsequent fragmentation of the $H_2 \cdots CO$ dimer (see Supplementary Note 3 for more details). Figure 1b presents the measured photoion-photoion coincidence (PIPICO) spectrum for the two-body Coulomb explosion channels, where the horizontal and vertical axes represent the time-of-flight (TOF) of the first and second ion, respectively. The selected signals of interest are plotted in Fig. 1c. Upon dissociative double ionization of the $H_2$ monomer $(H_2 + n\hbar\omega \rightarrow H^+ + H^+ + 2e^-)$ and CO monomer $(CO + n\hbar\omega \rightarrow C^+ + O^+ + 2e^-)$, $(H^+, H^+)$ and $(C^+, O^+)$ ion pairs are produced, respectively. The symbol $n$ is the number of absorbed photons of angular frequency $\omega$ of the laser pulse. Moreover, the ion yields arising from the bimolecular reaction of $H_2$ and CO are identified as the singly charged ion pairs $(H_2^+, CO^+)$ with a mass-to-charge ratio $m/q = 2$ and 28, and $(H^+, HCO^+)$ with $m/q = 1$ and 29, which are observed only when the molecular jet is precooled to produce $H_2 \cdots CO$ dimers. Among the two-body fragmentation channels of the $H_2 \cdots CO$ dimer, the branching ratio for the $(H_2^+, CO^+)$ and $(H^+, HCO^+)$ channel is ~91.4% and ~8.6%, respectively. The production of the formyl ion $HCO^+$ serves as a clear indicator of C−H bond formation in the bimolecular reaction as we confirmed using the tailored two-color pulse in the following.

Various reaction processes lead to the formation of different products in the laser-induced bimolecular reaction of $H_2$ and CO[17,39,40]. Upon laser irradiation, the neutral $H_2 \cdots CO$ dimer may undergo two-site double ionization, producing thereby the $(H_2^+, CO^+)$ ion pair. In this ion pair, $H_2^+$ can readily dissociate into the $(H^+, H)$ pair. Subsequently, the departing energetic neutral H can approach $CO^+$. The side at which H approaches $CO^+$ (approaching the C or O side) determines the subsequent interaction, leading to the formation of the $(H^+, HCO^+)$ or $(H^+, HOC^+)$ ion pair. Previous theoretical studies have demonstrated that the $H_2 \cdots CO$ dimer is most stable when two molecular axes are colinear[41–43]. For simplicity, we primarily consider the bimolecular reaction starting from two colinear configurations of the dimer: H-H$\cdots$C-O and H-H$\cdots$O-C. In this case, the emission direction of the dissociating pair of $H^+$ and H along the dimer axis, relative to the orientation of CO, allows one to distinguish between the two reaction scenarios involving the neutral H interacting with $CO^+$ from different sides. For instance, with a fixed orientation of CO tagged with the laser field pointing from C to O, the emission of H along or against the laser field direction would result in the formation of $HCO^+$ or $HOC^+$, respectively.

To determine the ion emission direction with respect to the CO orientation and the laser field direction in the observed bimolecular reaction channel, we employ a phase-controlled linearly polarized (along z-axis) two-color laser setup (790 nm & 395 nm). Ruled by the asymmetric profile of the molecular orbital, the CO monomer is more likely to be ionized when the laser field points from the C to O atom[44]. As a result, as shown in Fig. 2a, when the laser phase $\phi_L = 0$ or $\pi$, the $C^+$ ion in the Coulomb explosion channel of $(C^+, O^+)$ exhibits distinct upward $(\mathbf{p_z} > 0)$ and downward $(\mathbf{p_z} < 0)$ emissions. To quantify the directional emission of $C^+$, we introduce the asymmetry parameter $\beta = [N_{+z} - N_{-z}]/[N_{+z} + N_{-z}]$, where $N_{+z}$ and $N_{-z}$ stand for the probabilities of ion fragments emitting to +z and -z directions, respectively. As shown in Fig. 2c, the obtained $\phi_L$-dependent asymmetry of $C^+$ emission in the $(C^+, O^+)$ channel allows for the identification of the CO orientation upon photoionization and the assignment of the laboratory frame of the asymmetric optical field. It is worthwhile noting that the impulsive pre-orientation of the neutral CO molecule[45–47] plays a minor role in the observed directional ion emission in the $(C^+, O^+)$ channel

because of the small degree of molecular orientation induced by the two-color pulses.

After determining the absolute phase $\phi_L$ and the CO orientation, we proceed to inspect the phase dependence of ion emission in the $(H^+, HCO^+)$ channel. Figure 2b shows the measured $\phi_L$-dependent momentum distribution $\mathbf{p_z}$ of $H^+$ in the $(H^+, HCO^+)$ channel. Clearly visible are the phase modulation of $H^+$ emitting upward $(\mathbf{p_z} > 0)$ and downward $(\mathbf{p_z} < 0)$ with momentum $|\mathbf{p_z}|$ ~20 atomic units (a.u.). The corresponding $\phi_L$-dependent asymmetry map, depicting the directional emission of $H^+$ with an asymmetry amplitude of approximately 12%, is displayed in Fig. 2d. Interestingly, we observe that the $H^+$ is predominantly emitted in the -z (+z) direction when the two-color laser phase $\phi_L = 0$ ($\pi$). This observation indicates a preference for $H^+$ emission against the laser field direction. In accordance with momentum conservation, the emission of the accompanying $HCO^+$ exhibits a completely opposite phase dependence, as shown in Fig. 2f.

Since the formation of the $(H^+, HCO^+)$ ion pair is accompanied by the dissociation channel of $(H^+, H)$, that is

$$H_2 + n\hbar\omega \rightarrow H_2^+ + e^- \rightarrow H^+ + H + e^- \tag{1}$$

the comparison between the phase dependence of $H^+$ emitting from $(H^+, HCO^+)$ and that of $H^+$ from $(H^+, H)$ could tell us the approaching side of neutral H toward $CO^+$. We consider the emission of H (or $H^+$) with kinetic energy comparable with that of $H^+$ in the $(H^+, HCO^+)$ channel. Figure 2e shows the $\phi_L$-dependent emission asymmetry of $H^+$ measured from the $(H^+, H)$ channel with momentum $\mathbf{p_z} < 5$ a.u., where $H^+$ mostly emits against the laser field direction, albeit with a small phase offset compared to $H^+$ in the $(H^+, HCO^+)$ channel. The fragmentation process of singly ionized $H_2^+$ driven by a two-color laser field involves different dissociation pathways[48] resulting in varying dissociation times and laser phases experienced by the $H_2^+$ nuclear wave packet. This leads to different phase dependences of $H^+$ emission and thus the small phase offset of the asymmetric proton emission in the two channels. Consequently, we assign the emission direction of neutral H to be mainly along the laser field direction.

Given that the laser field direction has been tagged to be from the C atom to the O atom in the CO molecule, our experimental results indicate that H mostly emits towards the C side rather than the O side while approaching $CO^+$, thus favoring the formation of $HCO^+$ over $HOC^+$. The directional ion emission scenario suggests that the $H_2 \cdots CO$ dimer undergoing bimolecular reaction predominantly adopts the configuration with an arrangement of H-H$\cdots$C-O. This finding aligns with the observation of the kinetic energy release (KER) of the $(H_2^+, CO^+)$ channel, primarily formed via the Coulomb-exploded double ionization process by directly breaking the vdW bond of the $H_2 \cdots CO$ dimer. According to the Coulomb's law, the bond distance of ~7.77 a.u. between $H_2$ and CO extracted from the measured KER with a mean value of about 3.5 eV, is in excellent agreement with the predicted values of ~7.76 a.u., where the $H_2 \cdots CO$ dimer is expected to possess a linear geometry with the C atom pointing toward the $H_2$ molecule[49]. The observed preferential $\phi_L$-dependent ejection of $H^+$ and $HCO^+$ ions thus demonstrate the subfemtosecond manipulation of ion emission direction towards C−H bond formation in the bimolecular reaction.

## Real-time visualization of C−H bond formation dynamics

We now address another essential question: how fast does the bimolecular reaction proceed to form the C−H bond in the $HCO^+$ cation? To resolve the reaction dynamics in time, we perform pump-probe measurements using two linearly polarized 790 nm femtosecond laser pulses. Figure 3a displays the measured KER spectrum of the nuclear fragments in the $(H^+, HCO^+)$ channel as a function of the pump-probe time delay. Several interesting features are clearly visible from the

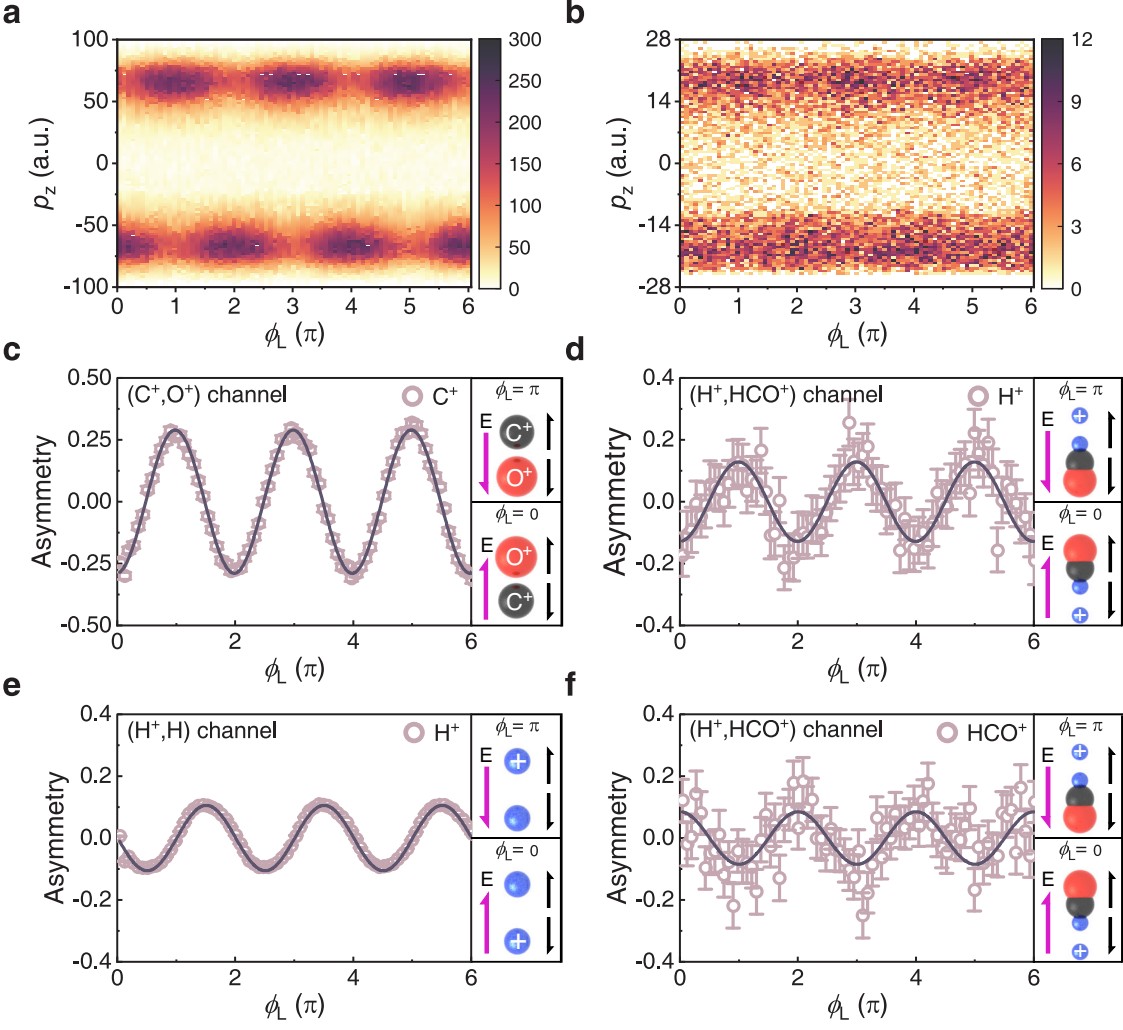

**Fig. 2 | Directional emission of ion fragments in the two-color scheme.**
**a**, **b** Phase-dependent momentum distribution $\mathbf{p_z}$ of **a** the C$^+$ emission in the (C$^+$, O$^+$) channel, and **b** H$^+$ emission in the (H$^+$, HCO$^+$) channel, as measured in the two-color experiments. **c**–**f** Asymmetry parameter of (**c**) C$^+$ emission in the (C$^+$, O$^+$) channel, (**d**) H$^+$ emission in the (H$^+$, HCO$^+$) channel, **e** H$^+$ emission in the (H$^+$, H) channel, and **f** HCO$^+$ emission in the (H$^+$, HCO$^+$) channel, as a function of the two-color relative phase $\phi_L$. The solid sinusoidal curves represent fits to the measured data by using $A(\phi_L) = A_0\sin[\pi(\phi_L - \phi_{offset})]$, where $A_0$ and $\phi_{offset}$ are the amplitude and phase offset of the asymmetry parameter, respectively. The fitting parameters for each curve can be found in the Supplementary Note 2. The error bars in **c**–**f** are defined

by $\sqrt{\frac{(Y_{+Z} - Y_{-Z})^2}{(Y_{+Z} + Y_{-Z})^3} + \frac{1}{(Y_{+Z} + Y_{-Z})}}$, where $Y_{+Z}$ and $Y_{-Z}$ represent the yield of ions with $\mathbf{p_z} > 0$ and $\mathbf{p_z} < 0$, respectively. The right panels of **c**–**f** schematically illustrate the field-direction-dependent directional emission of ion fragments in the different channels when the $\phi_L = 0$ (bottom) and $\phi_L = \pi$ (top). The pink arrows labeled with E and the black arrows are used to indicate the direction of the field maximum of the asymmetric two-color fields and the emission direction of the involved ion fragments, respectively. The blue sphere with and without plus sign in the panel of **e** indicate the ionic H$^+$ and neutral H, respectively.

data. The spectrum can be divided into two distinct regions with KER values above and below 2.5 eV, referred to as the high-KER and low-KER regions, respectively. The high-KER band, with a mean value of ~3.7 eV, is largely independent of the time delay, except for the appearance of a gap structure in the (H$^+$, HCO$^+$) yield around zero delay. The gap structure arises from the intense depletion of H$_2$··CO dimers and even the otherwise produced HCO$^+$ cations into other smaller fragments at high laser intensities when the pump and probe pulses temporally overlap. On the other hand, in the low-KER region, a gradual reduction in nuclear kinetic energy and an increase in the (H$^+$, HCO$^+$) yield are observed as the pump-probe delay increases in the positive delay range. These distinct features observed in the two energy regions indicate distinct reaction dynamics in the formation of the (H$^+$, HCO$^+$) channel.

In the high-KER channel, which is independent of the pump-probe time delay, the (H$^+$, HCO$^+$) ion pair is mainly formed through a double ionization process, which can be induced by either the pump or the

probe pulse individually. The reaction process can be denoted as

$$H_2 \cdot\cdot CO \xrightarrow{\text{pump/probe}} H_2^+ + CO^+ + 2e^- \rightarrow H^+ + HCO^+ + 2e^-. \quad (2)$$

The dissociation of H$_2^+$ and the subsequent emission of H towards CO$^+$ are strongly favored when the molecular axis of H$_2^+$ is parallel to the laser polarization. As a result, the angular distribution of the (H$^+$, HCO$^+$) channel exhibits a pronounced anisotropy along the laser polarization direction. Moreover, the high-KER spectrum can be obtained with a single pump or probe pulse, as displayed in the left panel of Fig. 3a. This suggests that a fast pathway for the formation of (H$^+$, HCO$^+$) is triggered via the double ionization process within a single pulse of a duration of ~36 fs.

For the low-KER channel, the obvious time dependence suggests that the (H$^+$, HCO$^+$) signal mainly stems from the joint contribution of the pump and probe pulses. Previous studies[42,50] suggest that the (H$^+$, HCO$^+$) formation can be attributed to an effective ion-neutral

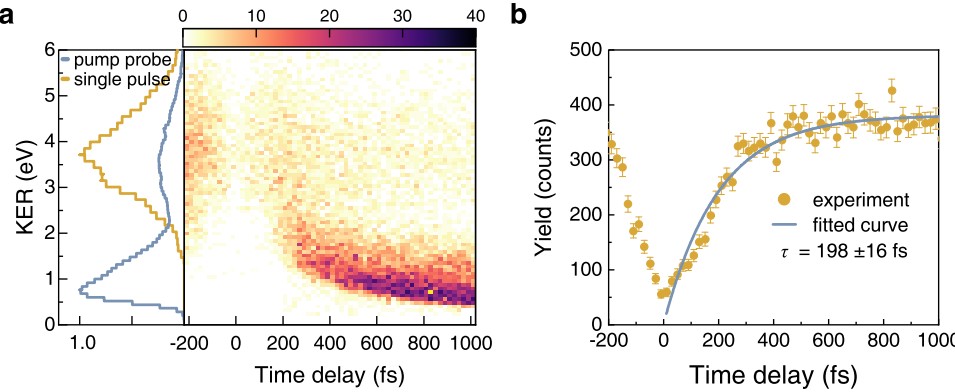

**Fig. 3 | Experimental results of the (H⁺, HCO⁺) channel in the pump-probe experiment.** **a** Right panel: measured yield of ion fragments in the (H⁺, HCO⁺) channel as a function of kinetic energy and pump-probe time delay. Left panel: normalized KER spectrum of nuclear fragments in the (H⁺, HCO⁺) channel obtained by integrating over all time delays in the pump-probe experiment (blue curve), and the results from a single linearly polarized laser pulse (orange curve). **b** Delay-dependent yield of ion fragments in the (H⁺, HCO⁺) channel. Corresponding exponential fit to the data points is shown by the blue solid curve. The error bars in the experimental results represent the statistical error derived from Poisson distribution, expressed as $\sqrt{N_{\mathrm{yield}}}$, where $N_{\mathrm{yield}}$ are the statistic total count of the measured ion fragments.

exothermic reaction. As induced by the pump and probe pulses, the reaction in the low-KER channel can be denoted as

$$H_2 \cdots CO \xrightarrow{\text{pump}} H_2 + CO^+ + e^- \rightarrow H + HCO^+ + e^- \xrightarrow{\text{probe}} H^+ + HCO^+ + 2e^-. \quad (3)$$

Upon photoionization of the CO molecule within the dimer by the pump pulse, a net attractive force emerges between $H_2$ and $CO^+$. As the neighboring neutral $H_2$ molecule approaches the $CO^+$ fragment, the H·H bond can be broken with a low barrier[42], leading to the formation of the (H, HCO⁺) pair. Subsequently, the outgoing neutral H atom can be ionized by the latter-arriving probe pulse during its separation from HCO⁺, resulting in the observation of the (H⁺, HCO⁺) ion pair with kinetic energy acquired from Coulomb explosion. As shown in Fig. 3a, the gradual decrease in kinetic energy with increasing time delay can be attributed to the expanding distance between H and HCO⁺ upon the probe-pulse-induced ionization of H into H⁺. The time-dependent yield of (H⁺, HCO⁺) indicates that the delayed probe pulse also plays a role in interrupting the reaction process for the formation of HCO⁺. As HCO⁺ is not yet stably formed, the system can undergo fragmentation into various many-body products.

To extract the formation time of HCO⁺ in the low-KER channel, we plotted in Fig. 3c the yield of (H⁺, HCO⁺) fragments as a function of pump-probe time delay. As the delay increases, HCO⁺ becomes more stable and its yield increases due to the relived interruption effect from the probe pulse. The presence of a plateau structure at large delays (> 600 fs) implies that HCO⁺ is almost stably formed, so that the probe pulse can no longer interrupt the formation process. To analyze the yield distribution to extract the time information, we fit the yield data using an exponential function $Y(t) = Y_0 + \alpha_0 \exp(-t/\tau)$, where $\alpha_0$ and $Y_0$ are the amplitude and offset of the (H⁺, HCO⁺) yield, and $\tau$ is the time constant. With a 94% confidence level for the fitting parameters, we extracted an effective formation time of HCO⁺ of approximately 198 ± 16 fs.

## Discussion

To gain further insights into the ultrafast dynamics of the $H_2 + CO^+$ reaction, we perform molecular dynamics simulations (see Methods section for details). For simplicity, we investigate the reaction starting from a $H_2 \cdots CO$ dimer with the most stable colinear configuration. Note that we also carried out simulations for other initial configurations, where the results show that the colinear configuration with $H_2$ at C side is the main configuration for the formation of HCO⁺ from $H_2$ and CO (Supplementary Note 6). By defining three reaction coordinates of $R_0$,

$R_1$, and $R_2$, the potential energy surfaces of the cationic $(H_2 \cdots CO)^+$, the cationic $(H_2 \cdots OC)^+$, and the neutral $H_2 \cdots CO$ dimer, sliced at $R_0 = 2.14$ a.u., are shown in Fig. 4a–c, respectively. The equilibrium geometry of the neutral $H_2 \cdots CO$ is determined to be situated at $R_1 = 7.09$ a.u., and $R_2 = 1.42$ a.u. Upon two-site double ionization, the distance between the two charge centers is calculated to be 7.8 a.u., in agreement with the measured KER. In contrast, the charge center for $H_2 \cdots OC$ is 9.0 a.u. apart upon ionization, deviating from experimental findings. This comparison implies that the neutral dimer is more likely to adopt the H·H··C·O configuration rather than H·H··O·C, leading to a prominent production of $(H_2 \cdots CO)^+$ over $(H_2 \cdots OC)^+$. Another rationale for the formation of the C−H bond instead of the O−H bond lies in the potential energy surface of $(H_2 \cdots OC)^+$ as shown in Fig. 4b, where a prominent reaction barrier suppresses the production of HOC⁺.

As depicted in Fig. 4a, c, the reaction dynamics for the formation of HCO⁺ can be understood as a three-step process. Upon Franck-Condon excitation, the cationic state $(H_2 \cdots CO)^+$ is initially populated around the equilibrium geometry of the neutral state (denoted as step 1). Subsequently, the nuclear dynamics evolves within the force field of the cationic state, with a typical trajectory depicted by the blue curve in Fig. 4a. Clearly, when the CO molecule is singly ionized, the $H_2$ molecule starts to approach the $CO^+$ ion, forming a transient state where all four particles are close (denoted as step 2). Eventually, the HCO⁺ ion is formed when a neutral H atom departs from other particles at longer time delays (denoted as step 3). During the formation of HCO⁺, one H atom approaches the neighboring $CO^+$ ion, accompanied by the departure of the other H atom simultaneously. By collecting the trajectories that lead to the formation of HCO⁺, we extracted the histogram of formation times, as shown in Fig. 4d. Through fitting the most probable formation time is determined to be ~201 fs, corresponding to the peak formation rate, which is in good agreement with the measured formation time of ~198 ± 16 fs.

In conclusion, we have demonstrated that the C−H bond can be formed via a bimolecular reaction of two small inorganic molecules of $H_2$ and CO within a van der Waals dimer driven by infrared femtosecond laser pulses. By manipulating the emission direction of the dissociating fragments using a tailored two-color laser field, the bimolecular reaction dynamics leading to the formation of the C−H bond and the associated HCO⁺ cation can be identified and controlled. Moreover, employing a pump-probe scheme, we precisely clocked the ultrafast formation time of HCO⁺ in different reaction pathways. In contrast to the faster double ionization pathway, the slower reaction pathway involving ion-neutral interactions takes about 198 ± 16 fs. Our molecular-level study of the C−H bond formation at low temperatures

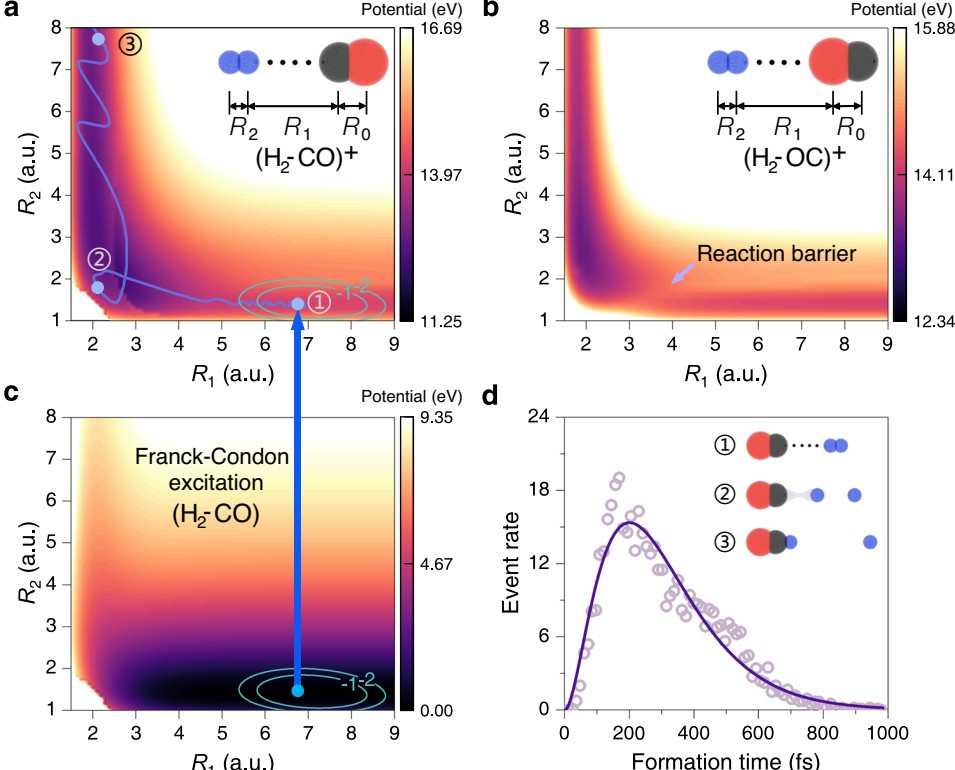

**Fig. 4 | Potential energy surfaces and HCO⁺ formation time from molecular dynamics simulation. a–c** Calculated potential energy surfaces of the **a** cationic $(H_2\cdots CO)^+$, **b** $(H_2\cdots OC)^+$, and **c** neutral $H_2\cdots CO$ dimer, sliced at $R_0 = 2.14$ a.u., with the ground state distribution marked by cyan contour lines. A typical reaction trajectory to form HCO⁺ is illustrated in panel **a** by the blue curve. **d** Histogram of the formation time of HCO⁺ obtained from the sampled reaction trajectories in the molecular dynamics simulations. The solid curve is the fitting to the data points. The inset illustrates the three-step reaction processes toward the formation of HCO⁺ as depicted in **a**.

casts light on the underlying light-induced inorganic bimolecular reaction processes, and carries substantial potential for comprehending the pivotal transition from inorganic to organic matter, thereby providing insights into the origins of the abundant organic compounds distributed throughout the universe. Our work also opens the opportunity towards the real-time tracking and coherent control of the stereodynamical bimolecular reaction dynamics[51] using waveform-manipulated light fields.

## Methods

### Experimental scheme

The experimental approach was based on the laser-driven reaction microscopy technique. As schematically illustrated in Supplementary Fig. 1 in the Supplementary Note 1, femtosecond laser pulses (25 fs, 790 nm, 10 kHz) produced from a multipass amplifier Ti:sapphire laser system were introduced into a two-color scheme and a pump-probe scheme for different experimental purposes.

In the two-color scheme, phase-controlled linearly polarized two-color laser pulses were produced in a colinear geometry[44] by frequency-doubling the femtosecond laser pulse in a 150 μm-thick $\beta$-barium borate ($\beta$-BBO) crystal. The polarization of the fundamental-wave (FW, 790 nm, y-polarized) was adjusted to align parallel to that of the second harmonic (SH, 395 nm, z-polarized) using a dual-wavelength wave plate. Birefringent $\alpha$-BBO crystals were used to adjust the rough time lag between the FW and SH pulses. A pair of fused-silica wedges installed on motorized linear positional delay stage was then employed to fine-tune the relative phase $\phi_L$ between FW and SH waves of the two-color pulses. The created electric field of the two-color laser pulses can be expressed as $E(t) = E_{FW}(t)\cos(\omega_{FW}t) + E_{SH}(t)\cos(\omega_{SH}t + \phi_L)$, where $E_{FW}(t)$ and $E_{SH}(t)$ represent the envelope, $\omega_{FW}$

and $\omega_{SH}$ denote the frequencies of the FW and SH fields, and $\phi_L$ is the relative phase between the FW and SH components. The intensities of FW and SH fields in the reaction region were estimated to be ~8 ×10¹³ W/cm² and ~1.4 × 10¹⁴ W/cm², respectively.

In the pump-probe scheme, the linearly polarized femtosecond laser pulse derived from the laser system was fed through a Mach-Zehnder interferometer to produce the pump and probe pulses. A motorized delay stage was utilized to precisely control the time delay between pump and probe pulses, scanning continuously from -220 fs to 1020 fs with a step size of 10 fs. The peak intensities of the pump and probe pulses in the interaction region were estimated to be ~1.6 × 10¹⁴ W/cm² and ~2 × 10¹⁴ W/cm², respectively. By tracing the autocorrelation signal in the time-dependent yield of singly ionized target molecules, the temporal duration of the pump and probe pulses were estimated to be ~36 fs (full width at half maximum).

The laser pulses in the different schemes were tightly focused by a concave mirror ($f = 75$ mm) inside the COLTRIMS chamber, where the laser pulses interacted with the supersonic molecular beam. A gas mixture of neutral $H_2$ (95%) and CO (5%) was expanded through a 5-μm-diameter nozzle with a driving pressure of approximately 18 bar. The ratio setting of the gas mixture ensures a relatively high proportion of the dimer-associated ion signal with respect to that from the monomer as induced by a single laser shot. The nozzle was precooled by connecting to a Dewars filled with liquid nitrogen. The resulting molecular beam consisted of $H_2$ and CO monomers, as well as $H_2\cdots CO$ dimers. Upon laser-induced reactions, the ionic fragments produced were accelerated by a weak homogenous electric field (~7 V/cm) towards a time- and position-sensitive multichannel plate detector located at the end of the spectrometer. The measured nuclear fragments ejected from the same molecular monomer or dimer can be unambiguously

identified based on the momentum conservation law. The three-dimensional momenta of the detected particles were retrieved through offline analysis of the measured times-of-flight (TOF) and positions of impact.

## Molecular dynamics simulation

Computational simulations were carried out to investigate the dynamics during the chemical reactions, yielding time-resolved insights into bond formation and cleavage processes. In agreement with the literatures[42,43,50], we found that the $H_2 \cdot \cdot CO$ dimer is most stable within a colinear configuration. To obtain an optimized structure of the colinear $H_2 \cdot \cdot CO$ dimer, we randomly placed the four atoms in a linear arrangement and relaxed the geometry using the OpenMolcas package[52]. Setting up the reaction coordinates $R_0$, $R_1$, and $R_2$ as defined in the insets of Fig. 4 of the main text, the three-dimensional potential energy hypersurfaces were calculated with the complete-active-space self-consistent-field (CASSCF) method at the CAS(12,6) and CAS(11,6) levels using the aug-cc-pVTZ basis set for the neutral and cationic dimer states, respectively. Upon photoionization of CO, a Franck-Condon excitation promoted the nuclear wavepacket from the neutral electronic state to the cationic state, and subsequent molecular dynamics was thereby initiated. The ground-state nuclear wave function on the neutral dimer state was used as the Franck-Condon wave packet (depicted by cyan contour lines), which was obtained from imaginary-time propagation by solving the time-dependent Schrödinger equation (TDSE). The initial positions for the subsequent molecular dynamics simulations were sampled then from the Franck-Condon wave packet, which naturally took account of anharmonic effects compared to traditional sampling methods. The initial momenta were set to zero. For sufficient statistics, we subsequently carried out molecular dynamics simulations with a swarm of 10000 nuclear trajectories using the classical-trajectory Monte Carlo (CTMC) method. The nuclear coordinates were then evolved on the potential energy hypersurface calculated by CASSCF, and those leading to the formation of $HCO^+$ are collected, as shown in Supplementary Fig. 3 in the Supplementary Note 4. We consider $HCO^+$ formed when the approaching distance $R_1$ is smaller than 4 a.u. and the departing distance $R_2$ is larger than 5 a.u.[35]. Subsequently, we analyze the formation time of the corresponding trajectories and a histogram of the formation time is constructed. The histogram is fitted with the Gamma distribution $f(t) = (t^{k-1}e^{-t/\delta})/(\Gamma(k)\tau^k)$ frequently used to model time distributions in Poisson processes, where $k = 3$ is the shape parameter, $\delta$ is the scale parameter, and $\Gamma(k)$ is the gamma function. With the fitting, the most probable formation time is determined to be ~201 fs, corresponding to the peak formation rate, which is in good agreement with the measured formation time of ~198 ± 16 fs.

## Data availability

The source data that support the main figures within this article are available from the Zenodo database[53].

## Code availability

All the codes that support the findings of this study are available from the corresponding author upon request.

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

## Acknowledgements

This work was supported by the National Natural Science Foundation of China (Grant Nos. 12227807 J.W., 12241407 J.W., 12304377 W.Z., and 92150105 H.N.), and the Science and Technology Commission of Shanghai Municipality (Grant No. 21ZR1420100 H.N. and 23JC1402000 J.W.), and the Shanghai Pujiang Program (23PJ1402600 W.Z.).

## Author contributions

Z.J., H.H., C.L., L.Z., S.P., J.Q., M.S., Z.Y., P.L., H.N., W.Z., and J.W. contributed to the experimental measurements and data analysis. H.H. and H.N. performed the simulations. Z.J., H.H., H.N., W.Z., and J.W. wrote the paper. All authors participated in the discussion of the results and commented on the manuscript.

## Competing interests

The authors declare no competing interests.
