## [Peer Review File · Nature Communications]

Ultrafast photoinduced C-H bond formation from two small inorganic moleculesREVIEWER COMMENTS

Reviewer #1 (Remarks to the Author):

The manuscript titled 'Ultrafast C-H photobonding from two small inorganic molecules' by Jiang et al. reported an interesting coherent control method on bimolecular reactions induced by ultrashort laser pulses. This experiment is an extension of the prior work, which focused on measuring the formation of trihydrogen cation from H₂ molecular dimers (as reported in Nat. Chem. 15 1224-1228 and Nat. Chem. 15, 1229-1235). However, I am pleased to see that the authors applied this technique to a heteromolecular dimer, H₂-CO, where they observed the formation of HCO⁺ from the bimolecular reaction H₂ + CO⁺ → HCO⁺ + H. Additionally, they demonstrated that this reaction can be coherently controlled using a two-color field. Employing a pump-probe scheme, they extracted the formation time of the C-H bond.

CO and H₂ are two fundamental molecules that are vital in the formation of numerous other molecules in the interstellar medium. Both the experimental results and the molecular dynamics simulation are convincing, and they agree well with each other. From this point of view, I think this paper holds general interest and should be suitable for publication in Nature Communications. However, before providing my full recommendation, I kindly request the authors to consider the following points

1. In the introduction, the authors provided an overview of previous studies on C-H bond formation using collision and scattering measurements and discussed the advantages of using a van der Waals molecular dimer. However, the concept of coherent control in bimolecular reactions was not adequately introduced. To enhance the reader's understanding, it is recommended to include information about coherent control of bond formation (e.g., as explored in J. Chem. Phys. 92, 1126–1131 (1990) and PRL 114, 233003 (2015)) and relevant references.
2. The production of H₂-CO dimer in COLTRIMS is challenging, as it requires a cold temperature of the nozzle and high pressure of the gas jet. I am wondering why 95% H₂ and 5% CO gases are mixed rather than a different ratio (e.g. 50% H₂ 50% CO). Would the (95% and 5%) ratio lead to the best efficiency for H₂-CO dimer generation?
3. The authors extracted various double ionization channels from the photoion-photoion coincidence spectrum. It remains unclear how the authors specifically chose the H⁺ + H channel, considering that neutral hydrogen is undetectable by the spectrometer. Clarification is needed regarding the selection process for different channels (single and double ionizations). Perhaps more details can be provided in this regard.
4. In Figure 2 c-f, the right (top-bottom) panels show the emission direction of the photoions relative to the asymmetric laser field (for illustration purposes). Unfortunately, these panels lack description in the main text or figure caption. Also, can the authors comment on the small phase offset (line 151) compared to H⁺ in the H⁺ + HCO⁺ channel? Is it related to a time delay between the two dissociation (H⁺ + H and H⁺ + HCO⁺) processes?
5. What is the branching ratio of HCO⁺ + H⁺ compared to other double ionization channels?
6. In the introduction (lines 70-72), it was mentioned that the fast double ionization can be triggered by a single pulse within 36 fs. This expression could be misleading, as readers might interpret the 36 fs as the formation time rather than the pulse duration. Since the high kinetic energy release is not dependent on the pump-probe delay, it might be more appropriate to omit the expression "within 36 fs."
7. For the slow reaction leading to the formation of HCO⁺, did the authors observe the H₂ roaming mechanism (as discussed in Nat. Comm. 9, 5186, (2018))? This could also be addressed through molecular dynamics simulation.
8. In lines 224-226, the authors mentioned that they investigated the reaction starting from an H₂-CO dimer with the most stable colinear configuration for simplicity. However, it would be instructive to provide insights into other configurations. Compared to the colinear configuration, what are the potential energies of other configurations? If these energies are not significantly higher, other configurations of H₂-CO dimers and zero-point energy should also be considered in the molecular dynamics simulations.

Reviewer #2 (Remarks to the Author):

The present work reports a detailed experimental study of a photo-induced bimolecular reaction using state-of-the-art methods. A fast pathway (about 36 fs) to the formation of HCO⁺ via double ionization of a H₂-CO van der Waals dimer as well as a slow pathway (about 198 fs) via single ionization are described. The slow pathway is also studied via molecular dynamics simulations. The authors write: "Our findings shed light on the most fundamental inorganic bimolecular reaction occurring in the interstellar clouds."

With the present form of the manuscript, the authors do not convince me that their results are sufficiently groundbreaking to warrant publication in Nature Communications. To that end, I have several comments, suggestions, and questions:

- 1) The study of photo-induced bimolecular reactions, starting with a van der Waals dimer, started many years ago. This is not reflected in the list of references where the pioneering works are missing. In particular, the works of C. Wittig and co-workers (e.g., JCP 1985) and the time-resolved pump-probe experiments by A.H. Zewail and co-workers (JCP 1987) starting from the (HI)-(CO₂) van der Waals complex. Furthermore, the authors refer to very recent work (Refs. 37 and 38) on similar photo-induced bimolecular reactions starting from a van der Waals dimer.
- 2) Several processes and ionic species are mentioned throughout the text. A more systematic notation is required. For example, for the single ionization, "low-KER channel", the reactions should be written such that all particles appear in the reaction scheme:
$$\text{H}_2\text{-CO} + h\nu \rightarrow (\text{H}_2\text{-CO})^+ + e^- \rightarrow \text{HCO}^+ + \text{H} + e^-$$
A similar notation should be used in Line 188 for the double ionization, "high-KER channel". Furthermore, for the pump-probe measurements, the reaction schemes of pump and probe pulses should be clearly separated. The notation in Line 145/146: "...dissociation of (H⁺, H)..." is not clear. I suppose that the authors refer to the dissociation of H₂⁺ ?
- 3) Are the conditions of intense infrared irradiation relevant for bimolecular reactions in interstellar clouds?
- 4) The ionization energy in the first step is around 14 eV (according to Fig. 4 a and c) which is much higher than the photon energies of the laser pulse. What is the mechanism, multiphoton or tunneling ionization?
- 5) The authors do not clearly describe that the two-color laser pulse induces molecular orientation (appropriate references are missing!) in addition to ionization. Is the frequency distribution independent of the relative phase ϕ_L ? The authors use the phrase "coherent control". Is this justified? Genuine coherent control is control via phases with a fixed frequency distribution.
- 6) In the description of the molecular dynamics simulations, it is stated that: "The initial positions ... were sampled then from the Franck-Condon wave packet ...". What about the initial momenta? Furthermore, it is suggested (line 330) that Wigner sampling cannot take anharmonic effects into account (with a reference to a paper that describes the Wigner function for an anharmonic Morse oscillator!). This is not correct.

To conclude, my overall impression is that the authors fail to convince the reader that the level of novelty is sufficiently high to warrant publication in Nature Communications.

Reviewer #3 (Remarks to the Author):

Review For "Ultrafast C-H photobonding from two small inorganic molecules."

In this study, Z. Jiang, et al. used ultrafast pump-probe spectroscopy on van der Waals dimers of H₂-CO to find the time scale of C-H bond formation, with a slow (within 36 fs) and fast pathway (198 ± 16 fs) to form the C-H bond. They also demonstrate how changing the laser phase affects the direction of H⁺ and HCO⁺ emission. The results support the conclusions and claims.

I am surprised by the term "photobonding" used in the title when the experiment is based on ionization followed the bimolecular reaction between H₂ and CO, which occurs after the laser pulse is over.

Major Notes

1. Given the focus of this manuscript is on a bimolecular reaction starting from a van der Waals dimer, it is surprising that the pioneering work done by Zewail, DOI: 10.1063/1.453280 is not cited. That work focuses on the bimolecular reaction of H + CO₂ starting from HI--CO₂ dimers using femtosecond laser pulses.
2. In the manuscript, the information about the fitting parameters used for the sinusoidal fits used in Figure 2, exponential rise fit (except the tau) in Figure 3, or Gamma distribution used in Figure 4 is not present. Please add the parameters for the fits in the manuscript or the SI.
3. For Figure 3b, the time zero feature is not shown. Show a full time-resolved curve that goes from negative time delay to positive time delay.
4. In Figure 2e, in the first ¼ n, explain why there is a gap in asymmetry that is not present in the rest of the phase.
5. When discussing molecular dynamics simulations, there is only discussion of the slow pathway of formation and in the histogram, there are several trajectories that take place in under 100 fs. What percent of the trajectories undergo the fast or slow pathway?

Minor Notes

2. Figure 1 caption has two periods at the end of line 532.
3. Line 82: "well defined" to "well-defined"

Reviewer #4 (Remarks to the Author):

I have reviewed article NCOMMS-23-42576-T entitled "Ultrafast C-H photobonding from two small inorganic molecules" by Zhejun Jiang et al., submitted to Nature Communication.

The manuscript describes the formation of C-H bonds and in particular, the time scale and molecular level control, which they claim remained elusive. The team investigated the light-induced bimolecular reaction using a van der Waals dimer composed of two molecules, H₂ and CO. The pump-probe technique used a two-color photons scheme with a reaction microscope or COLTRIMS to identify the pathways leading to C-H photobonding and producing HCO⁺. The team reports an ultrafast formation time of 198 fs +/- 16 fs.

The paper is well written and the literature is cited sufficiently but the results are not noteworthy. A very important figure is missing, the one that shows the production of HCO⁺.

It is unclear to this reviewer what Fig. 1b shows. It certainly doesn't show the photoion-photoion (PIPICO) coincidences; ie channels from the two-body CE resulting from H₂ and CO. The authors need to provide the PIPICO map with all the channels that result from the ionization and formation of new fragments. Such a map should unambiguously show channels formed from these ions H⁺, H₂⁺, HCO⁺, CO⁺. For example the authors need to provide the PIPICO for these channels (C⁺,O⁺); (H⁺, HCO⁺).

Fig.3 claims to show the dynamics of the (H⁺,HCO⁺); ie., the KER as a function of time delay but this reviewer will need to see first the PIPICO of the channel in question.

This manuscript is not suitable for publication in Nature communication because formation of CH bond is not new and measuring its time dynamics is not sufficient. I would recommend that this paper be submitted instead to J. Chem. Phys. where there is already a body of work regarding this research as cited by the authors. I would also recommend that the authors include the PIPICO map of all the channels in their revised manuscript to be submitted elsewhere as other referees will require it.

Minor remarks:

Typo, like 33, the word thereinto, doesn't belong in the sentence.

Author Response to Reviewer's Comments:

We thank all the Reviewers for their detailed reading of our manuscript and their thoughtful comments. We have revised the manuscript according to their suggestions. In this response letter, we reproduce the original *comments by the Reviewers in italics and black*, provide *our responses in blue* and the *changes to the manuscript in red*.

Reply to Reviewer #1

The manuscript titled 'Ultrafast C-H photobonding from two small inorganic molecules' by Jiang et al. reported an interesting coherent control method on bimolecular reactions induced by ultrashort laser pulses. This experiment is an extension of the prior work, which focused on measuring the formation of trihydrogen cation from H₂ molecular dimers (as reported in Nat. Chem. 15 1224-1228 and Nat. Chem. 15, 1229-1235). However, I am pleased to see that the authors applied this technique to a heteromolecular dimer, H₂-CO, where they observed the formation of HCO⁺ from the bimolecular reaction H₂ + CO⁺ → HCO⁺ + H. Additionally, they demonstrated that this reaction can be coherently controlled using a two-color field. Employing a pump-probe scheme, they extracted the formation time of the C-H bond.

CO and H₂ are two fundamental molecules that are vital in the formation of numerous other molecules in the interstellar medium. Both the experimental results and the molecular dynamics simulation are convincing, and they agree well with each other. From this point of view, I think this paper holds general interest and should be suitable for publication in Nature Communications. However, before providing my full recommendation, I kindly request the authors to consider the following points.

Reply: We very much appreciate the Reviewer for these very positive remarks on our work and the recommendation for publication in Nature Communications. We have responded point by point to the Reviewer's comments and improved our manuscript by closely following the Reviewer's suggestions as detailed in the following.

Comment #1. *In the introduction, the authors provided an overview of previous studies on C-H bond formation using collision and scattering measurements and discussed the advantages of using a van der Waals molecular dimer. However, the concept of coherent control in bimolecular reactions was not adequately introduced. To enhance the reader's understanding, it is recommended to include information about coherent control of bond formation (e.g., as explored in J. Chem. Phys. 92, 1126–1131 (1990) and PRL 114, 233003 (2015)) and relevant references.*

Reply: We thank the Reviewer for this comment concerning the concept of coherent control. The coherent control of chemical reaction relies on the ability to precisely manipulate the quantum interference of the phase and amplitude of the wavefunction of the reacting molecules, resulting in the coherent control of product yields or reaction pathways. This can be achieved by using the light sources, such as lasers, which possess coherent properties. By tailoring the laser parameters (including intensity, pulse duration, frequency, polarization, phase, and etc.) to match the specific energy levels and dynamics of the molecules through coherent excitation, the bond formation and cleavage in the chemical reactions can be controlled in a desired way. As mentioned by Reviewer,

for instance, in ref. [J. Chem. Phys. 92, 1126–1131 (1990)], it is theoretically proposed that the tunable time delay in a pulse-pulse scheme can be employed to control the branch ratio of bimolecular reaction pathways. In ref. [Phys. Rev. Lett. 114, 233003 (2015)], a femtosecond laser pulse is shaped by tuning the chirp parameters to enable the coherent control of bond making in a photoinduced bimolecular chemical reactions.

In our work, the coherent control of the bimolecular reaction in $\text{H}_2\cdot\text{CO}$ dimer is realized by employing a phase-controlled two-color laser field. It has been demonstrated that the phase-controlled two-color laser field can be used to achieve the coherent control of the directional bond breaking in the unimolecular fragmentation reaction (see e.g., refs. Phys. Rev. Lett. 71, 692(1993), Phys. Rev. Lett. 74, 4799(1995), Phys. Rev. Lett. 103, 223201 (2009), Phys. Rev. A 96, 033405 (2017)). The underlying mechanism lies in the control of intramolecular localization of bound electron, which can be quantum mechanically understood as the result of coherent superposition of the dissociated nuclear wave packet of same final kinetic energy but opposite parities. In our case, the asymmetric laser fields created by tuning the relative phase of the two-color pulses is employed to control the directional emission of ions from bond breaking of H_2 molecules and the subsequent formation of HCO^+ with a controlled emission directionality. Therefore, the control over of the C-H bond formation dynamics in the bimolecular reaction is achieved.

To make it clear to readers, by following the Reviewer's suggestions, in the revised manuscript, we have added the following sentences to introduce the concept of coherent control, where the mentioned refs. J. Chem. Phys. 92, 1126–1131 (1990), Phys. Rev. Lett. 114, 233003 (2015) and other relevant references are included.

“Moreover, the incoherent population of scattering states in collision measurements presents a significant obstacle in achieving coherent control of bond cleavage and formation in bimolecular reactions. To realize the coherent control, the precise manipulation of quantum interference of phase and amplitude of the wavefunctions of the reacting molecules is required²⁸⁻³².” (line 65-69 on page 3)

Comment #2. *The production of $\text{H}_2\text{-CO}$ dimer in COLTRIMS is challenging, as it requires a cold temperature of the nozzle and high pressure of the gas jet. I am wondering why 95% H_2 and 5% CO gases are mixed rather other a different ratio (e.g. 50% H_2 50% CO). Would the (95% and 5%) ratio lead to the best efficiency for $\text{H}_2\text{-CO}$ dimer generation?*

Reply: In our experiment, the strong laser pulses interact with both the dimer molecules, $\text{H}_2\cdot\text{H}_2$, $\text{CO}\cdot\text{CO}$, and $\text{H}_2\cdot\text{CO}$, and the monomer molecules, H_2 and CO , at the interaction region. Since the ionization potential of CO (14.07 eV) is much lower than that of H_2 (15.6 eV), for a given laser intensity, the yields of laser-created ion signals from CO monomer will be significantly larger than that from H_2 monomer. In fact, we have tested in the experiment that when the gas source is mixed with 50% H_2 and 50% CO , the detected ion yields was overwhelmed by the signals from CO monomer, which in term largely reduces the proportion of detected signal from the bimolecular reaction of $\text{H}_2\cdot\text{CO}$ dimer. The (95% and 5%) ratio might not be the best efficiency for $\text{H}_2\cdot\text{CO}$ dimer generation, but in our experiment such setting can significantly improve the proportion of dimer-associated ion signal with respect to the large amount of signal from the monomer as induced by a single laser shot.

To make it clear to readers, in the revised manuscript, we have added the following sentences in the Method section.

“The ratio setting of the gas mixture ensures a relatively high proportion of the dimer-associated ion signal with respect to that from the monomer as induced by a single laser shot.” (line 330-332 on page 13)

Comment #3. *The authors extracted various double ionization channels from the photoion-photoion coincidence spectrum. It remains unclear how the authors specifically chose the H⁺ + H channel, considering that neutral hydrogen is undetectable by the spectrometer. Clarification is needed regarding the selection process for different channels (single and double ionizations). Perhaps more details can be provided in this regard.*

Reply: We thank the Reviewer for this comment, which mainly concerns the methods for the identification of different fragmentation channels. For the two-body Coulomb-exploded double ionization channels, such as (H⁺, H⁺), (C⁺, O⁺), (H₂⁺, CO⁺), and (H⁺, HCO⁺), the ion pairs in different channels can be unambiguously identified using the photoion-photoion coincidence (PIPICO) spectrum, which shows the time-of-flight (TOF) spectra of the first and second ions in the horizontal and vertical axes, respectively. The coincidence measurement of the two-body Coulomb explosion events constrains the corresponding ion signal in the diagonal lines of the TOF coincidence map. By imposing the recoil momentum rule of $p_{\text{sum}} = |p_{\text{ion1}} + p_{\text{ion2}}| < 4$ a.u. for the sum-momentum of coincidentally detected two ion fragments, the false coincidence signal in the Coulomb-exploded double ionization channels can be significantly suppressed. For the interested dissociative single ionization channel of (H, H⁺), as mentioned by the Reviewer, the neutral hydrogen (H) atom is undetectable by the spectrometer. The events associated to the (H, H⁺) channel are thus identified according to the detection of H⁺ ion. Typically, the recoil momentum of H⁺ ion from the (H, H⁺) channel as accessed via bond softening mechanism is below 15 a.u., which is much lower than that from the (H⁺, H⁺) channel (>20 a.u.). Therefore, a momentum gate of $p_{\text{H}^+} = \sqrt{p_{\text{ionx}}^2 + p_{\text{iony}}^2 + p_{\text{ionz}}^2} < 15$ a.u. is employed for the momentum distribution of H⁺ to clearly distinguish the H⁺ ion signal in (H, H⁺) channel from that in (H⁺, H⁺) channel, where p_{ionx} , p_{iony} , p_{ionz} are the ion momentum distributions along x, y, and z direction, respectively.

To make it clear to readers, by following the suggestion of the Reviewer, we have added more details on the identification of different fragmentation channels in the Supplemental Materials.

“In the data analysis, the involved two-body Coulomb-exploded double ionization channels are unambiguously identified by using the photoion-photoion coincidence (PIPICO) spectrum. As shown in Fig. 1b in the main text, the signals of ion pair originating from the two-body breakup of the same molecular entity are constrained along the sharp diagonal lines of the time-of-flight correlation map. Several two-body fragmentation channels can be identified. The (H⁺, H⁺) and (C⁺, O⁺) channels are produced from the Coulomb-exploded double ionization of H₂ and CO monomers, respectively. The direct Coulomb explosion of H₂·H₂, CO·CO, H₂·CO dimer induced by two-site double ionization contributes to the (H₂⁺, H₂⁺), (CO⁺, CO⁺), and (H₂⁺, CO⁺) channel, respectively. Besides these ion pairs, the (H⁺, HCO⁺) with mass-to-charge ratio (m/q) of ion pair being m/q = 1 and 29 is also identified, which originate from the laser-induced molecular reactions of the H₂·CO

dimer. Note that the false coincidence fragmentation events and background ionization events are also recorded in the PIPICO spectrum. For instance, the vertical and horizontal sharp lines in the PIPICO spectrum with $m/q = 1, 2$ and 28 are the ion fragments of H^+ , H_2^+ , and CO^+ from the false coincidence fragmentation events. The recoil momentum rule of $p_{\text{sum}} = |p_{\text{ion1}} + p_{\text{ion2}}| < 4$ a.u. is imposed for the sum-momentum of coincidentally detected two ion fragments to suppress the false coincidence signal. The discrimination of the detected H^+ ion signals from the dissociative single ionization channel (H, H^+) and double ionization channel (H^+, H^+) relies on the different H^+ ion recoil momentum in the two channels. Typically, the recoil momentum of H^+ ion from the (H, H^+) channel as accessed via bond softening mechanism is below 15 a.u., which is relatively lower than that from the (H^+, H^+) channel (>20 a.u.). Therefore, a momentum gate of $p_{H^+} = \sqrt{p_{\text{ionx}}^2 + p_{\text{iony}}^2 + p_{\text{ionz}}^2} < 15$ a.u. is employed for the momentum distribution of H^+ to clearly distinguish the H^+ ion signal in (H, H^+) channel from that in (H^+, H^+) channel, where $p_{\text{ionx}}, p_{\text{iony}}, p_{\text{ionz}}$ are the ion momentum distributions along x, y, and z direction, respectively.” (line 39-64 on page 2-3 of the Supplemental Materials)

Comment #4. In Figures 2c-f, the right (top-bottom) panels show the emission direction of the photoions relative to the asymmetric laser field (for illustration purposes). Unfortunately, these panels lack description in the main text or figure caption. Also, can the authors comment on the small phase offset (line 151) compared to H^+ in the $H^+ + HCO^+$ channel? Is it related to a time delay between the two dissociation ($H^+ + H$ and $H^+ + HCO^+$) processes?

Reply: We thank the Reviewer for this comment, which concerns two aspects, 1) the description of the right panels in Figs. 2c-f, and 2) the small phase offset of the H^+ emission in (H^+, H) and (H^+, HCO^+) channels. We answer them separately in the following.

1) The right panels of Figs. 2c-f. are used to illustrate the emission direction of ion fragments in different fragmentation channels with respect to the direction of the field maximum of the asymmetric two-color fields when the relative phase $\phi_L = \pi$ (top panel) and $\phi_L = 0$ (bottom panel). In each panel, the pink arrow labeled with E is used to indicate the field direction of the two-color pulses. The emission direction of the ion fragments is indicated by the black arrows. Based on the orientation-dependent ionization and dissociation of CO molecule, i.e., the scenario that CO monomer is favored to be ionized when the laser field points from C to O atom, the direction of field maximum of the asymmetric two-color fields can be determined by inspecting the asymmetric emission of C^+ ion in (C^+, O^+) channel. As shown in Fig. 2a, when the $\phi_L = 0$, the preferable downward emission of C^+ (i.e., upward emission of O^+) indicates a field direction pointing from C (bottom) to O (top). It is the opposite case for $\phi_L = \pi$. Therefore, in Fig. 2c, the identified field direction and the corresponding emission direction of C^+ and O^+ for $\phi_L = 0$ and $\phi_L = \pi$ are illustrated in the bottom and top panel, respectively. After determining the field direction at $\phi_L = 0$ and $\phi_L = \pi$, the emission of H^+ and H fragments in (H, H^+) channel, and the emission of H^+ and HCO^+ fragments in (H^+, HCO^+) channel are illustrated in the bottom and top panel in Figs. 2e, d and f, respectively, according to the measured results.

To make the panel illustration clearer to readers, as suggested by the Reviewer, we have slightly modified the panels and added the following sentences in the caption of Fig. 2 for detailed description.

“The right panels of c-f schematically illustrate the field-direction-dependent directional emission of ion fragments in the different channels when the $\phi_L = 0$ (bottom) and $\phi_L = \pi$ (top). The pink arrows labeled with E and the black arrows are used to indicate the direction of the field maximum of the asymmetric two-color fields and the emission direction of the involved ion fragments, respectively. The blue sphere with and without plus sign in the panel of e indicate the ionic H^+ and neutral H, respectively.”

2) Regarding the small phase offset of the directional H^+ emission in (H, H^+) channel (Fig. 2c) compared to the H^+ emission in (H^+, HCO^+) channel (Fig. 2d), the different laser-driven fragmentation processes of singly ionized H_2^+ in the two channels as accessed via various dissociation pathways might be responsible for the observations. The asymmetric emission of H^+ in (H, H^+) channel has been generally understood by the scenario of electron localization among the two dissociating nuclei, which requires superposition of nuclear wave packets (NWP) with the same kinetic energy release (KER) but opposite parties. When driven by a two-color laser field, the dissociation of H_2^+ NWP can be accessed via different pathways along the potential surface of $1s\sigma_g$ and $2p\sigma_u$ state, including $1\omega_{SH}-1\omega_{FW}$, $1\omega_{FW}$, $net-2\omega_{FW}$, $1\omega_{SH}$, $1\omega_{SH}+2\omega_{FW}-1\omega_{FW}$, and $3\omega_{FW}$ pathways, where the ω_{FW} and ω_{SH} represent the photon energy of the fundamental wave and second harmonics, respectively (see ref. [Phys. Rev. A 96, 033405 (2017)] for pathway definition). For the H^+ emission within a certain KER region, the different dissociation times of the involved pathways and various laser phases experienced by the NWP during the dissociation processes would result in different phase dependences. According to the measured KER distributions, the observed phase-dependence of H^+ emission in Fig. 2c is assigned to be mostly contributed from the interference of the $1\omega_{SH}-1\omega_{FW}$, and $1\omega_{FW}$ pathways. Since the dissociation process of H_2^+ in $H_2\cdot CO$ dimer system might be influenced by the altered potential surfaces, the H^+ emission in (H^+, HCO^+) channel accessed via dissociation pathways differing from that in (H, H^+) channel could lead to the small phase offset in the phase-dependent asymmetric proton emission.

To further clarify this point, we have added/rewritten the following sentences in the revised manuscript.

“The fragmentation process of singly ionized H_2^+ driven by a two-color laser field involves different dissociation pathways⁴⁸, resulting in varying dissociation times and laser phases experienced by the H_2^+ nuclear wave packet. This leads to different phase dependences of H^+ emission and thus the small phase offset of the asymmetric proton emission in the two channels.” (line 172-176 on page 6-7)

Comment #5. *What is the branching ratio of $HCO^+ + H^+$ compared to other double ionization channels?*

Reply: As shown in Table R1, we extracted the measured yields of the involved double ionization channels in the pump-probe experiments, including the (H^+, H^+) , (C^+, O^+) , (H_2^+, CO^+) and (H^+, HCO^+) channels. The (H^+, H^+) and (C^+, O^+) channels are generated from the double ionization of H_2 and CO monomer, where the yield of (H^+, H^+) channel is dominant. Due to the relatively small proportion of the formed $H_2\cdot CO$ dimer compared to the monomer in the gas jet, the yields of (H_2^+, CO^+) and (H^+, HCO^+) channels are much less than that of (H^+, H^+) and (C^+, O^+) channel. Considering that both the (H_2^+, CO^+) and (H^+, HCO^+) channels are derived from the dissociative double ionization of $H_2\cdot CO$ dimer, the branch ratios of the two channels with respect to their sum yields are calculated to

be ~91.4% and ~8.6% for the (H_2^+ , CO^+) and (H^+ , HCO^+) channel, respectively.

Table R1. The yield of different double ionization channels measurement

	Yield (counts)	Ratio
(H_2^+ , CO^+)	192642	91.4%
(H^+ , HCO^+)	18115	8.6%
(C^+ , O^+)	581178	
(H^+ , H^+)	2.24365×10^7	

To make it clear to readers, we have added/rewritten the following sentences in the revised manuscript.

“Among the two-body fragmentation channels of the H_2 -CO dimer, the branching ratio for the (H_2^+ , CO^+) and (H^+ , HCO^+) channel is ~91.4% and ~8.6%, respectively.” (line 113-115 on page 4-5)

Comment #6. *In the introduction (lines 70-72), it was mentioned that the fast double ionization can be triggered by a single pulse within 36 fs. This expression could be misleading, as readers might interpret the 36 fs as the formation time rather than the pulse duration. Since the high kinetic energy release is not dependent on the pump-probe delay, it might be more appropriate to omit the expression "within 36 fs."*

Reply: We thank the Reviewer for noticing us this point. Here, we want to state that the fast pathway for the formation of C-H bond could be triggered by a single pulse, which means that the two-site double ionization of H_2 -CO dimer is completed within a single pulse of a duration of 36 fs, resulting the consequent bimolecular reaction process in forming the C-H bond. As suggested by the Reviewer, in the revised manuscript, we have rephrased the mentioned sentence as following.

“The fast pathway, accessed via a two-site double ionization of the dimer, can be triggered by a single pulse.” (lines 77-78 on page3)

Comment #7. *For the slow reaction leading to the formation of HCO^+ , did the authors observe the H_2 roaming mechanism (as discussed in Nat. Comm. 9, 5186, (2018))? This could also be addressed through molecular dynamics simulation.*

Reply: We thank the Reviewer for this comment. In ref. [Nat. Comm. 9, 5186 (2018)] as mentioned by the Reviewer, the roaming of formed neutral H_2 and later combination with a terminal proton of ionized methanol/ethanol molecules leads to the formation of H_3^+ ion. In our case, the H_2 roaming mechanism indicates that the H_2 molecule within the $\text{H}_2 \cdots \text{CO}$ dimer migrates from the O side to C side or from C side to O side. Our analyses have shown that the slow pathway leading to the formation of HCO^+ is accessed via a reaction process involving single ionization of CO molecule within the $\text{H}_2 \cdots \text{CO}$ dimer accompanied with the bond breaking of neutral H_2 molecule. The dissociation of neutral H_2 molecule occurring at C side or O side will result in the formation of HCO^+ or HOC^+ , respectively. Therefore, for the slow pathway leading to the formation of HCO^+ , the H_2 roaming mechanism, if existing, indicates a migration of neutral H_2 from O side to C side of CO molecule.

Fig. R1 Simulated evolution of nuclear coordinates (a) R_1 , (b) R_2 , (c) θ in the $(\text{H}_2\cdot\text{CO})^+$ dimer. The distance of the C-O bond is fixed at 1.1565 \AA , R_1 is the vector pointing from the midpoint of the C-O bond to $\text{H}^{(1)}$, R_2 is the vector pointing from $\text{H}^{(1)}$ to $\text{H}^{(2)}$, which always keeps the same direction as R_1 in this configuration. θ is the angle between the R_1 vector and the C-O bond.

To address such scenario, we performed molecular dynamics simulations. The results suggest that the yield of H_2 roaming from the O side to the C side leading to the formation of HCO^+ is far less compared to the regular HCO^+ formation pathway. We have evolved one thousand trajectories on the potential energy surface (PES) of cationic $(\text{H}_2\cdot\text{CO})^+$ in Jacobi coordinates (θ, R_1, R_2) shown in Fig. R1 with the initial orientation angle $\theta = -\pi$ and the initial position (R_1, R_2) sampled by the ground-state nuclear wave function for the PES $(\theta = -\pi, R_1, R_2)$. The time evolution of the trajectories in Fig. R1(a)-(b) shows that in most cases the two molecules have moved away from each other within 200 fs, as their intermolecular distance R_1 increases with time and the H-H bond length R_2 remains stable without breaking. Figure R1(c) shows that the change of the orientation angle θ of most trajectories is less than 0.1π over 200 fs, which means that H_2 molecule have moved away from CO molecule well before it roams from O side to C side. Therefore, the contribution of the H_2 roaming mechanism to the formation of HCO^+ is negligible.

To make it clear to readers, we have included such studies in the section “5. Roaming mechanism” in the Supplemental Materials.

Comment #8. In lines 224-226, the authors mentioned that they investigated the reaction starting from an $\text{H}_2\text{-CO}$ dimer with the most stable colinear configuration for simplicity. However, it would be instructive to provide insights into other configurations. Compared to the colinear configuration, what are the potential energies of other configurations? If these energies are not significantly

higher, other configurations of H₂-CO dimers and zero-point energy should also be considered in the molecular dynamics simulations.

Reply: We thank the Reviewer for this comment. The potential energies of the neutral H₂-CO dimer in different configurations have been calculated in ref. [J. Chem. Phys. 139, 164315 (2013)]. As shown in Fig. R2(a), the geometry of H₂·CO can be described naturally using the Jacobi coordinates ($R, \theta_1, \theta_2, \phi, r_{\text{CO}}, r_{\text{H}_2}$). There, R is a vector pointing from the centre of mass of CO to the centre of mass of H₂, θ_1 is the angle between R and a vector pointing from atom O to C, θ_2 is the angle between R and a vector pointing from H⁽²⁾ atom to H⁽¹⁾, ϕ the dihedral angle between the two planes defined by vector of R with the CO molecule and with the H₂ molecule, r_{CO} is the bond length of the CO molecule and r_{H_2} is the bond length of the H₂ molecule.

Fig. R2 (a) Complete Jacobi coordinates for H₂-CO. (b) Vibrational averaged interaction energy curves for neutral H₂-CO at various relative orientations. Figure adapted from ref. [J. Chem. Phys. 139, 164315 (2013)].

The potential energy for H₂-CO at different relative orientations can be expressed as $\Delta V(\theta_1, \theta_2, \phi)$, as shown in Figs. R2(b) and R2(c). Of these curves, the red curve ($\theta_1=0^\circ, \theta_2=0^\circ, \phi=0^\circ/90^\circ$) corresponding to the colinear configuration has much lower energies than other curves, meaning that most of the H₂·CO dimer would exist colinearly with H₂ at the C side. Furthermore, the H₂·CO dimer in other configurations with different relative orientations need to undergo relative intermolecular rotation after single ionization to form HCO⁺, which would make the HCO⁺ yield much lower than the colinear configuration.

Fig. R3 Simulated evolution of nuclear coordinates (a) R_1 , (b) R_2 in a $(\text{H}_2\text{-CO})^+$ dimer for a vertical configuration with H_2 at O side initially. R_1 is the vector pointing from C atom to the center of mass of H_2 , which is always colinear with the C-O bond orientation in this configuration. R_2 is the nuclear distance of H_2 molecule. θ is the relative orientation of H_2 and CO molecules.

Except the most stable initial colinear configuration as indicated by the red curve in Fig. R2, we have performed molecular dynamics simulations starting additionally from the second most stable vertical configuration with H_2 at the O side corresponding to the black curve ($\theta_1=180^\circ$, $\theta_2=90^\circ$, $\phi=0^\circ/90^\circ$) in Figs. R2(b) and (c). In such case, the rotation and stretching of the H-H bond are needed for H_2 to react with CO to form HCO^+ . In the simulations, we set up the Jacobi coordinates (θ , R_1 , R_2) as defined in the inset of Fig. R3(a). We subsequently evolved 10000 trajectories with the initial orientation angle $\theta = 90^\circ$ and the initial position sampled by the ground-state nuclear wave function for PES ($\theta = 90^\circ$, R_1 , R_2). Figures R3(a) and R3(b) show the evolution of all trajectories in R_1 and R_2 coordinates over time. Most trajectories reflect that the two molecules are moving away from each other with the absolute value of R_1 increasing and R_2 remaining stable, meaning that little HCO^+ can be formed in this situation.

In conclusion, the colinear configuration with H_2 at C side is the main configuration for the formation of HCO^+ from H_2 and CO, for its high population due to the lowest potential energy and the high yield owing to the formation process without a reaction barrier or any necessity of relative rotation.

To make it clear to readers, we have added the following sentences in the revised manuscript and included the detailed studies in the section “6. Other initial configurations” in the Supplemental Materials.

“Note that we also carried out simulations for other initial configurations, where the results show that the colinear configuration with H_2 at C side is the main configuration for the formation of HCO^+ ”

from H₂ and CO (see Supplemental Materials).” (lines 250-253 on page 9)

In summary, thanks again for the Reviewer’s helpful comments and suggestions on our manuscript. We hope our detailed response to Reviewer’s comments and the accordingly revised new version of manuscript can convinces the Reviewer that our manuscript is now appropriate for publication in Nature Communications.

Reply to Reviewer #2

The present work reports a detailed experimental study of a photo-induced bimolecular reaction using state-of-the-art methods. A fast pathway (about 36 fs) to the formation of HCO⁺ via double ionization of a H₂-CO van der Waals dimer as well as a slow pathway (about 198 fs) via single ionization are described. The slow pathway is also studied via molecular dynamics simulations. The authors write: "Our findings shed light on the most fundamental inorganic bimolecular reaction occurring in the interstellar clouds."

With the present form of the manuscript, the authors do not convince me that their results are sufficiently groundbreaking to warrant publication in Nature Communications. To that end, I have several comments, suggestions, and questions:

Reply: We thank the Reviewer for the careful review of our work. To make it more convincing that the molecular-level study of C-H bond formation in our work appeals to the broad readership of Nature Communications, we have improved our manuscript by closely following the Reviewer's suggestions as detailed in the following.

Comment #1. *The study of photo-induced bimolecular reactions, starting with a van der Waals dimer, started many years ago. This is not reflected in the list of references where the pioneering works are missing. In particular, the works of C. Wittig and co-workers (e.g., JCP 1985) and the time-resolved pump-probe experiments by A.H. Zewail and co-workers (JCP 1987) starting from the (HI)-(CO₂) van der Waals complex. Furthermore, the authors refer to very recent work (Refs. 37 and 38) on similar photo-induced bimolecular reactions starting from a van der Waals dimer.*

Reply: We sincerely appreciate the Reviewer for this valuable comment. We very much appreciate the kind notice of the Reviewer on the using of van der Waals (vdW) dimer system to study the photo-induced bimolecular reaction. To make it clear to readers, by following the Reviewer's suggestion, in the revised manuscript, we added/rewrote the following sentences, where the mentioned pioneering works are included in the references list.

"Unlike bimolecular reactions in the beam scattering experiments, where the intermolecular distance is uncertain, the two interacting molecules within a molecular dimer system has a well-defined equilibrium geometry. Consequently, as pioneered by Wittig's³³ and Zewail's³⁴ works, the starting point of the light-driven bimolecular reaction dynamics in a vdW dimer is well-defined and can be precisely tracked^{35,36}." (line 86-91 on page 4)

Comment #2. *Several processes and ionic species are mentioned throughout the text. A more systematic notation is required. For example, for the single ionization, "low-KER channel", the reactions should be written such that all particles appear in the reaction scheme: H₂-CO + hv → (H₂-CO)⁺ + e⁻ → HCO⁺ + H + e⁻. A similar notation should be used in Line 188 for the double ionization, "high-KER channel". Furthermore, for the pump-probe measurements, the reaction schemes of pump and probe pulses should be clearly separated. The notation in Line 145/146: "...dissociation of (H⁺, H)..." is not clear. I suppose that the authors refer to the dissociation of H₂⁺?*

Reply: We thank the Reviewer for noticing us the points regarding the systematic notation for the investigated reaction channels. As suggested by the Reviewer, in the revised manuscript, we have

rewritten the sentences with notations for both the low-KER and high-KER channel of C-H bond formation reaction.

“In the high-KER channel, which is independent of the pump-probe time delay, the (H^+ , HCO^+) ion pair is mainly formed through a double ionization process, which can be induced by either the pump or the probe pulse individually. The reaction process can be denoted as $H_2 \cdot CO \xrightarrow{\text{pump/probe}} H_2^+ + CO^+ + 2e^- \rightarrow H^+ + H + CO^+ + 2e^- \rightarrow H^+ + HCO^+ + 2e^-$ ” (line 208-212 on page 8)

“As induced by the pump and probe pulses, the reaction in the low-KER channel can be denoted as $H_2 \cdot CO \xrightarrow{\text{pump}} H_2 + CO^+ + e^- \rightarrow H + HCO^+ + e^- \xrightarrow{\text{probe}} H^+ + HCO^+ + 2e^-$.” (line 222-224 on page 8)

We also added the notations for (H^+ , H^+) and (C^+ , O^+) channels, where (H^+ , H^+) channel refers to the dissociative double ionization of H_2 molecule, respectively, and (C^+ , O^+) channel refers to the dissociative double ionization of CO molecules. The (H^+ , H) channel in our experiment refers to the dissociative single ionization of H_2 molecule, which can be denoted as $H_2 + n\hbar\omega \rightarrow H_2^+ + e^- \rightarrow H^+ + H$. The following sentences are added/rewrote in the revised manuscript:

“Upon dissociative double ionization of the H_2 monomer ($H_2 + n\hbar\omega \rightarrow H^+ + H^+ + 2e^-$) and CO monomer ($CO + n\hbar\omega \rightarrow C^+ + O^+ + 2e^-$), (H^+ , H^+) and (C^+ , O^+) ion pairs are produced, respectively. The symbol n is the number of absorbed photons of angular frequency ω of the laser pulse.” (line 106-109 on page 4)

“Since the formation of the (H^+ , HCO^+) ion pair is accompanied by the dissociation channel of (H^+ , H), that is, $H_2 + n\hbar\omega \rightarrow H_2^+ + e^- \rightarrow H^+ + H + e^-$, ...” (line 162-163 on page 6)

Comment #3. *Are the conditions of intense infrared irradiation relevant for bimolecular reactions in interstellar clouds?*

Reply: We thank the Reviewer for this comment. In interstellar media, the unimolecular or bimolecular reactions forming the interstellar molecular species could be mostly initiated by the ultraviolet light radiation or cosmic radiation, which consists high energy charged particles and x-rays. To investigate the interstellar chemical reactions under laboratory conditions, researchers used the electron impact, proton impact, highly-charged ion collision, and the femtosecond laser light as excitation sources to study the reaction mechanism. For instance, in ref. [Nat. Commun. **9**, 5186 (2018)], intense near-infrared laser pulses were used to study the formation of interstellar H_3^+ ion in the fragmentation of organic molecules.

The main goal of our work is to understand the molecular-level mechanism of the C-H bond formation from two small inorganic molecules. The using of intense near-infrared laser light as irradiation source can provide femtosecond time resolution for the time-resolved studies and allows us to achieve the control over the bimolecular reaction dynamics.

Comment #4. *The ionization energy in the first step is around 14 eV (according to Figs. 4 a and c) which is much higher than the photon energies of the laser pulse. What is the mechanism, multiphoton or tunneling ionization?*

Reply: We answer this question by referring to the Keldysh parameter $\gamma = \sqrt{\frac{I_p}{2U_p}}$, where I_p is the ionization potential, and $U_p = \sqrt{\frac{E^2}{4\omega^2}}$ with laser intensity of E and frequency of ω is the pondermotive energy. According to the Keldysh theory (Sov. Phys. JETP 20, 1307(1965)), when the $\gamma \gg 1$, the ionization is governed by the multiphoton ionization, while for $\gamma \ll 1$, the tunneling ionization is dominant. In the pump-probe experiment, the ionization of H₂-CO dimer can be induced by both the pump and probe pulses, with peak intensity of $\sim 1.6 \times 10^{14}$ W/cm² and $\sim 2 \times 10^{14}$ W/cm², respectively. With the ionization potential of H₂-CO dimer to be ~ 14.14 eV, the corresponding Keldysh parameter are calculated to be ~ 0.86 and ~ 0.76 , respectively, indicating a dominated tunneling ionization mechanism for the ionization of H₂-CO dimer in our study.

Comment #5. *The authors do not clearly describe that the two-color laser pulse induces molecular orientation (appropriate references are missing!) in addition to ionization. Is the frequency distribution independent of the relative phase ϕ_L ? The authors use the phrase “coherent control”. Is this justified? Genuine coherent control is control via phases with a fixed frequency distribution.*

Reply: We thank the Reviewer for this comment, which concerns two aspects: 1) the molecular orientation of CO molecule in the two-color scheme, and 2) the method of coherent control in our work. We answer them separately in the following.

1) We fully agree with the Reviewer that the two-color laser pulses can impulsively induce the molecular orientation of CO molecules in addition to the ionization. It has been demonstrated in previous studies (refs. Phys. Rev. Lett. 103, 153002(2009), Erratum Phys. Rev. Lett. 112, 159902 (2014), Phys. Rev. Lett. 104, 213901(2010), Phys. Rev. Lett. 109, 113001(2012), Phys. Rev. Letts. 112, 113005 (2014)) that the all-optical and field-free orientation of CO molecule can be achieved by using two-color laser fields. In general, in the experiment, a pump-probe scheme is required to characterize the degree of molecular orientation, where the CO orientation is induced by a two-color pump pulse and a delayed probe pulse is used to Coulomb-explode the molecule. For instance, as shown in Fig. R4 (adapted from ref. Phys. Rev. Lett. 103, 153002(2009)), the degree of CO orientation is characterized by the angle θ between the momentum of emitted C²⁺ ion fragments and the laser polarization direction. The phase-dependent orientation of CO molecules can be reflected by the distinct distributions of orientation parameter $\langle \cos\theta \rangle$ as a function of the pump-probe delay at $\phi_L = 0$ and $\phi_L = \pi$, where the orientation is most significant at the revival of the alignment, i.e., $t=8.7$ ps.

However, such impulsive pre-orientation of CO molecule by the two-color fields plays a minor role in here-observed phase-dependent directional ion emission in the dissociative ionization of CO molecules because of two reasons. First, the orientation degree is too small to affect the asymmetric ionization step. As shown in Fig. R4, the maximum orientation degree at the revival ($t=8.7$ ps) is about 0.06, which corresponds to an orientation angle $\theta \sim 86^\circ$. Considering the lower combined intensity of the two-color pulses used in our work (fundamental wave: $\sim 8 \times 10^{13}$ W/cm² and second harmonics: $\sim 1.4 \times 10^{14}$ W/cm²) compared to that used in ref. Phys. Rev. Lett. 103, 153002(2009) (fundamental wave: $\sim 1.3 \times 10^{14}$ W/cm² and second harmonics: $\sim 1.3 \times 10^{14}$ W/cm²), the orientation degree of CO molecule by the two-color pulses in our work would be even smaller (ref. Phys. Rev. Lett. 109, 113001(2012), Phys. Rev. Letts. 112, 113005 (2014)). Note that in our work

Editorial Note: Fig. R4 below is adapted with permission from De, S. et al. Field-Free Orientation of CO Molecules by Femtosecond Two-Color Laser Fields. *Phys. Rev. Lett.* **103**,153002 (2009). <https://doi.org/10.1103/PhysRevLett.103.153002>. ©2009 American Physical Society

a delayed probe pulse is required in the two-color scheme to characterize the orientation degree. Second, one can see in Fig. R4 that the maximum degree of orientation occurs at a large delay, for instance, at the revival of alignment ($t=8.7\text{ps}$). As reported in our pump-probe experiment, the formation time of C-H bond in the bimolecular reaction is measured to be around 200 fs. This indicates that the bimolecular reaction completes before the significant CO orientation appears, further indicating the minor role of pre-orientation of CO in the reaction dynamics.

To make it clear to readers, in the revised manuscript, we have added/rewrote the following sentences to refer to the molecular orientation of CO molecule, where appropriate references are included in the reference list.

“It is worthwhile noting that the impulsive pre-orientation of the neutral CO molecule⁴⁵⁻⁴⁷ plays a minor role in the observed directional ion emission in the (C^+ , O^+) channel because of the small degree of molecular orientation induced by the two-color pulses.” (line 146-149 on page 6)

Fig. R4 Orientation parameter $\langle \cos \theta \rangle$ as a function of pump-probe time delay for two opposite phase of the two-color pump pulse. Adapted from Fig. 2(b) in S. De et al., Phys. Rev. Lett. 103, 153002(2009), Erratum Phys. Rev. Lett. 112, 159902 (2014).

2) In our work, the coherent control of photoinduced bimolecular reaction dynamics is realized by employing a two-color scheme. The electric field of a phase-controlled linear two-color pulse can be expressed as $E(t) = E_{\omega}(t)\cos(\omega t) + E_{2\omega}(t)\cos(2\omega t + \phi_L)$, where $E_{\omega}(t)$ and $E_{2\omega}(t)$ represent the electric field amplitude profiles of the pulses with frequency of ω and 2ω , respectively, and ϕ_L is the relative phase between the two frequency components. As shown in Fig. R5, the waveform of the two-color laser fields can be controlled by tuning the relative phase ϕ_L . For instance, the direction of asymmetric maximal optical field can be tailored to point along ‘up’ or ‘down’ direction when $\phi_L=0$ or $\phi_L=\pi$. The phase-controlled two-color pulses have been employed to realize the coherent control of directional bond breaking in unimolecular photodissociation, where the directionality of the ion emission can be controlled by tuning the relative phase ϕ_L (please also see our response to the comment #1 by Reviewer #1). Here, a similar two-color phase control approach is employed to control over the bimolecular reaction dynamics by manipulating the ion emission direction in forming the C-H bond in HCO^+ ion. As concerned by the Reviewer, for each two-color laser pulse, the frequency distribution of the two-color components will not change as the relative phase ϕ_L changes, ensuring the coherence property of the laser pulse in the control scheme.

Fig. R5 Electric field distributions of the linearly polarized two-color (center wavelengths: 395nm & 790nm) laser fields for relative phase **a**, $\phi_L=0$, **b**, $\phi_L=0.5\pi$, **c**, $\phi_L=\pi$.

Comment #6. In the description of the molecular dynamics simulations, it is stated that: “The initial positions ... were sampled then from the Franck-Condon wave packet ...”. What about the initial momenta? Furthermore, it is suggested (line 330) that Wigner sampling cannot take anharmonic effects into account (with a reference to a paper that describes the Wigner function for an anharmonic Morse oscillator!). This is not correct.

Reply: We thank the Reviewer for pointing out the inappropriateness of the citation here. In our simulation, we use the ground-state nuclear wave function of the neutral $\text{H}_2\cdot\text{CO}$ dimer to sample the initial positions and momenta of the trajectories. As the phase S of the calculated wave function is flat, we set the initial momenta of all trajectories to zero according to $\mathbf{p} = \Delta S$, see, for example, refs. [Phys. Rev. Lett. 110, 243001 (2013), Phys. Rev. Lett. 117, 023002 (2016)]. This sampling method can naturally take account of the anharmonic effect. The citation was intended to stress the importance of the anharmonic effects. We agree with the Reviewer that the cited paper treats the anharmonic Morse oscillator. In the revised manuscript, we have rewritten this sentence without the citation:

“The initial positions for the subsequent molecular dynamics simulations were sampled then from the Franck-Condon wave packet, which naturally took account of anharmonic effects compared to traditional sampling methods. The initial momenta were set to zero.” (line 356-359 on page 14)

To conclude, my overall impression is that the authors fail to convince the reader that the level of novelty is sufficiently high to warrant publication in Nature Communications.

We appreciate the Reviewer for his/her feedback on our manuscript. We would like to further clarify the novelty of our work for the Reviewer in the following.

As is known, the carbon-hydrogen (C-H) bond plays a crucial role in organic compounds and is widely recognized to be responsible for existence of life in the universe. Exploring the formation of C-H bonds through reactions involving small inorganic molecules offers valuable insights into the transition from inorganic to organic matter and establishes connections between interstellar chemical networks and the origins of life. However, previous studies on the C-H bond formation have predominantly relied on full intermolecular collision and scattering measurements, which lack a definition of the spatiotemporal starting point for time-resolved studies. The detailed mechanism and dynamics of C-H bond formation, particularly the timescales and molecular-level coherent control of the dynamics, remain elusive to date.

In our work, we performed the molecular-level study of the C-H bond formation dynamics in a light-induced bimolecular reaction of a vdW dimer containing two small inorganic molecules, H₂ and CO, where the dimer is formed at a very low temperature condition. This reaction is the most fundamental inorganic bimolecular reaction occurring in the interstellar clouds that leads to the production of organic matter. As compared to the beam scattering studies, the C-H bond formation is initiated by laser-induced bimolecular reaction starting from a molecular dimer, which ensures a well-defined spatiotemporal starting point for time-resolved studies and coherent control of the reactions. As mentioned by the Reviewer, the approach for light-induced bimolecular reaction studies using vdW dimer system was pioneered by Zewail's and Wittig's group. Here, we show that such approach allows us to study the ultrafast dynamics and the control over of C-H bond formation at a molecular level.

The highlighted points of the results in our work includes: 1) By employing a tailored two-color laser field, the pathway leading to C-H bond making thereby producing HCO⁺ ions is clearly identified; 2) The coherent control of the bimolecular reaction process with directional emission of ionic resultants is achieved by tuning the relative phase of the two-color laser fields; 3) The ultrafast dynamics of photoinduced C-H bond making is characterized by performing femtosecond pump-probe measurements, where a formation time of 198 fs is unambiguously discriminated. Our molecular-level study of C-H bond formation at low temperatures would cast light on the fundamental light-induced bimolecular reaction processes occurring in cold planetary or interstellar gas clouds and thus the origin of the abundant organic compounds found throughout the universe.

To make the clarification of the above-mentioned point regarding the novelty of our work clear to readers, we have improved the abstract and introduction by highlighting the significance and novelty of our studies, where the following sentences were added/rewrote in the revised manuscript:

“Our findings shed light on the most fundamental inorganic bimolecular reaction occurring in the interstellar clouds, paving the route towards real-time visualization and coherent control over the dynamics with unprecedented precision.” (abstract)

“The investigation of C-H bond formation from the reaction of small inorganic substances holds particular significance as it provides profound insights into the fundamental transition from inorganic to organic matter and establishes connections between interstellar chemical networks and the emergence of life^{10,11}. Realizing coherent control of such C-H bond making process represents a significant advancement in the field of chemical organic synthesis.” (line 34-39 on page 2 in the introduction part)

In summary, thanks again for the Reviewer's helpful comments and suggestions on our manuscript. We hope our detailed response to Reviewer's comments and the accordingly revised new version of manuscript can convinces the Reviewer that our manuscript is now appropriate for publication in Nature Communications.

Reply to Reviewer #3

Review For “Ultrafast C-H photobonding from two small inorganic molecules.”

In this study, Z. Jiang, et al. used ultrafast pump-probe spectroscopy on van der Waals dimers of H₂-CO to find the time scale of C-H bond formation, with a slow (within 36 fs) and fast pathway (198 ± 16 fs) to form the C-H bond. They also demonstrate how changing the laser phase affects the direction of H⁺ and HCO⁺ emission. The results support the conclusions and claims.

Reply: We thank the Reviewer for the careful review and the accurate summary of our work. We have responded point by point to the Reviewer’s comments and improved our manuscript by closely following the Reviewer’s suggestions as detailed in the following.

I am surprised by the term “photobonding” used in the title when the experiment is based on ionization followed the bimolecular reaction between H₂ and CO, which occurs after the laser pulse is over.

Reply: We thank the Reviewer for this comment. In our study, the bimolecular reaction between H₂ and CO molecules is triggered by an initial single or double ionization of H₂-CO dimer as induced by the strong laser pulses. Therefore, we attribute the formation of C-H bond in the bimolecular reaction to a photoinduced process.

To avoid unnecessary confusion to readers, by following the Reviewer’s suggestion, in the revised new manuscript, we change the title of our manuscript to be “Ultrafast photoinduced C-H bond making from two small inorganic molecules”.

Major Notes

Comment# 1. *Given the focus of this manuscript is on a bimolecular reaction starting from a van der Waals dimer, it is surprising that the pioneering work done by Zewail, DOI: 10.1063/1.453280 is not cited. That work focuses on the bimolecular reaction of H + CO₂ starting from HI--CO₂ dimers using femtosecond laser pulses.*

Reply: We are grateful to the Reviewer for noticing us to refer to the pioneering works of the bimolecular reaction studies with van der Waals (vdW) dimer, which is also commented by Reviewer #2 (please also see our response to the Comment #1 by Reviewer #2). As compared to the work pioneered by Zewail’s group and Wittig’s group, we employed a vdW dimer system consisting two small H₂ and CO molecule to study the C-H bond formation from the simplest inorganic bimolecular reaction. Benefiting from the well-defined spatiotemporal starting point of the bimolecular reaction, the dynamics of the bimolecular reaction can be precisely tracked and the coherent control of the process is achieved by using waveform-controlled laser fields.

To make it clear to readers, by following the suggestion of the Reviewer #2 and #3, we have added references for the mentioned pioneering works and accordingly revised the following sentences in the revised manuscript.

“Unlike bimolecular reactions in the beam scattering experiments, where the intermolecular distance is uncertain, the two interacting molecules within a molecular dimer system has a well-defined equilibrium geometry. Consequently, as pioneered by Wittig’s³³ and Zewail’s³⁴ works, the

starting point of the light-driven bimolecular reaction dynamics in a vdW dimer is well-defined and can be precisely tracked^{35,36}." (line 86-91 on page 4)

Comment #2. *In the manuscript, the information about the fitting parameters used for the sinusoidal fits used in Figure 2, exponential rise fit (except the tau) in Figure 3, or Gamma distribution used in Figure 4 is not present. Please add the parameters for the fits in the manuscript or the SI.*

Reply: We thank the Reviewer for this comment. By following the Reviewer's suggestion, we have added the information of corresponding fitting parameters in the different fitting functions of the mentioned 1) sinusoidal fit, 2) exponential fit, and 3) Gamma distribution fit in the Supplemental Materials and rewrote/added the relevant sentences in the revised manuscript.

1) For the fitting parameters of sinusoidal curves in Figs. 2c-2f, we added the following sentences in the caption of Fig. 2 in the main text,

"The solid sinusoidal curves represent fits to the measured data by using $A(\phi) = A_0 \sin[\pi(\phi - \phi_{\text{offset}})]$, where A_0 and ϕ_{offset} are the amplitude and phase offset of the asymmetry parameter, respectively. The fitting parameters for each curve can be found in the Supplemental Materials." (line 575-579 on page 21)

and the corresponding parameter information in the section "2. The fitting parameters" in the Supplemental Materials.

"In Figs. 2c-2f of the main text, the distributions of the phase-dependent asymmetry amplitude are fitted using sinusoidal function $A(\phi) = A_0 \sin[\pi(\phi - \phi_{\text{offset}})]$, where A_0 and ϕ_{offset} are the amplitude and phase offset of the asymmetry parameter, respectively. The fitting parameters for Figs. 2c-2f are extracted as following: (1) Fig. 2c for C⁺ emission in (C⁺, O⁺) channel: $A_0 = 0.28 \pm 0.002$, $\phi_{\text{offset}} = 0.48 \pm 0.002$; (2) Fig. 2d for the HCO⁺ emission in (HCO⁺, H⁺) channel: $A_0 = 0.12 \pm 0.008$, $\phi_{\text{offset}} = 0.49 \pm 0.022$; (3) Fig. 2e for H⁺ emission in (H⁺, H) channel: $A_0 = 0.10 \pm 0.001$, $\phi_{\text{offset}} = 0.8 \pm 0.004$; (2) Fig. 2f: $A_0 = 0.08 \pm 0.011$, $\phi_{\text{offset}} = 1.50 \pm 0.041$." (line 21-28 on page 2 of the Supplemental Materials)

2) For the fitting parameters of the exponential yield increasing in Fig. 3b, we rewrote the following sentence in the revised main text,

"To analyze the yield distribution to extract the time information, we fit the yield data using an exponential function $Y(t) = Y_0 + \alpha_0 \exp(-t/\tau)$, where α_0 and Y_0 are the amplitude and offset of the (H⁺, HCO⁺) yield, and τ is the time constant." (line 242-245 on page 9)

and the corresponding parameter information in the section "2. The fitting parameters" in the Supplemental Materials.

"In Fig. 3b of the main text, the KER-integrated yield distributions of (HCO⁺, H⁺) channel is fitted by using an exponential function $Y(t) = Y_0 + \alpha_0 \exp(-t/\tau)$, where α_0 and Y_0 are the amplitude and offset of the (H⁺, HCO⁺) yield, and τ is the time constant. In the fitting, the parameters $\alpha_0 = -379.0 \pm 15.0$, $Y_0 = 381.0 \pm 6.3$, $\tau = 198 \pm 16.7$ are used." (line 29-32 on page 2 of the Supplemental Materials)

3) For the fitting parameters of the Gamma distribution in Fig. 4d, we added the following information in the Supplemental Materials.

“In Fig. 4d of the main text, the histogram shows the formation time distributions of (HCO^+ , H^+) channel extracted in the molecular dynamics simulations. The histogram is fitted by using the Gamma distribution $f(t)=(t^{k-1}e^{-t/\delta})/(\Gamma(k)\tau^k)$, where $k=3$ is the shape parameter, δ is the scale parameter, and $\Gamma(k)$ is the gamma function $\int_0^{+\infty} y^{k-1}e^{-y}dy$. In the fitting, the used parameters $\delta = 100.89$ fs.” (line 33-37 on page 2 of the Supplemental Materials)

Comment #3. For Figure 3b, the time zero feature is not shown. Show a full time-resolved curve that goes from negative time delay to positive time delay.

Reply: We thank the Reviewer for this comment. The Fig. 3b shows the KER-integrated yield of ion fragments in the (H^+ , HCO^+) channel as a function of the pump-probe time delay. To make it clear to reader, as suggested by the Reviewer, in the revised manuscript, we have modified the Fig. 3b by presenting the time-resolved yield data points from negative time delay to positive time delay. The revised Fig. 3b is also shown in Fig. R6 for a more convenient review.

Fig. R6. The modified Fig. 3b in the revised manuscript, showing the measured time-resolved yield data points from negative time delay to positive time delay. The solid blue curve is the exponential fit to the data point in the positive delay.

Comment #4. In Figure 2e, in the first $\frac{1}{4}\pi$, explain why there is a gap in asymmetry that is not present in the rest of the phase.

Reply: The small drop of asymmetry amplitude value at the data point of $\phi_L=0.06\pi$ in the first $\frac{1}{4}\pi$ of the phase-dependent asymmetry amplitude distribution shown in Fig. 2e is mainly caused by the mechanical problems of the motorized translational delay stage. In the phase-scan experiment, the delay stage was moved back-and-forth and scanned continuously. Since there is a bad point on the translational axis of the motor roller, when the delay stage passes this point, the communication between the stage and the data acquisition system will be interfered, which messes the counts of acquired data events at this phase step and thus the calculated value of asymmetry amplitude. The messy data point at the phase step of $\phi_L=0.06\pi$ should generally appear in the data distributions of all the detected channels. In fact, the small drop can also be observed

in Fig. 3c. However, as compared to the data points of (H⁺, H) channel shown in Fig. 3e, due to the less statistics and larger error bar of (C⁺, O⁺), and (HCO⁺, H⁺) channels, the small drop feature is less significant in Figs. 3c, 3d and 3f.

Comment #5. *When discussing molecular dynamics simulations, there is only discussion of the slow pathway of formation and in the histogram, there are several trajectories that take place in under 100 fs. What percent of the trajectories undergo the fast or slow pathway?*

Reply: We thank the Reviewer for this comment. In our work, all the calculated trajectories in the molecular dynamics simulations belong to the slow pathway. The fast and slow pathways are named to distinguish between two different HCO⁺ formation processes, based on their relative

reaction time. The two reactions are written as $\text{H}_2\cdot\text{CO} \xrightarrow{\text{pump/probe}} \text{H}_2^+ + \text{CO}^+ + 2e^- \rightarrow \text{H}^+ + \text{H} + \text{CO}^+ + 2e^- \rightarrow \text{H}^+ + \text{HCO}^+ + 2e^-$ (fast) and $\text{H}_2\cdot\text{CO} \xrightarrow{\text{pump}} \text{H}_2 + \text{CO}^+ + e^- \rightarrow \text{H} + \text{HCO}^+ + e^- \xrightarrow{\text{probe}} \text{H}^+ + \text{HCO}^+ + 2e^-$ (slow). Compared with the double ionization process of the fast pathway, we pay more attention to the sequential single ionization process of the slow pathway, which can be better time-resolved by a pump-probe process to extract reaction time information.

Correspondingly, in our simulation, we only simulated the formation process of HCO⁺ of the slow pathway. Firstly we calculated the potential energy surface (PES) of the neutral H₂·CO dimer and cationic (H₂·CO)⁺ under the specified reaction coordinates. Sampled by the ground-state nuclear wave function calculated by solving the time-dependent Schrödinger equation (TDSE) on the PES of the neutral H₂·CO dimer, we obtained the initial positions of all trajectories. Then, we evolve these trajectories on the PES of cationic (H₂·CO)⁺ with zero initial momentum to simulate the formation process of HCO⁺ from H₂ and CO⁺ after the neutral dimer single-ionized by the pump laser.

The reason for several trajectories taking place in under 100 fs is that the initial intermolecular distance of some trajectories is close enough, leading to the two molecules quickly coming together and completing the reaction within 100 fs. We would like to stress that even this is relatively fast, it still corresponds to the slow pathway because we categorize based on the formation mechanism of HCO⁺.

Minor Notes

1. *Figure 1 caption has two periods at the end of line 532.*

2. *Line 82: “well defined” to “well-defined”*

Reply: We thank the Reviewer for the very careful checks. We have made the corrections to the mentioned points in the revised manuscript.

In summary, we very much appreciate the Reviewer for his/her helpful comments. We have elaborated our observations and interpretations in this reply letter and accordingly improved the manuscript by closely following the suggestions of the Reviewer. We hope that this new submitted version of manuscript is now appropriate for publication in Nature Communications.

Reply to Reviewer #4

I have reviewed article NCOMMS-23-42576-T entitled "Ultrafast C-H photobonding from two small inorganic molecules" by Zhejun Jiang et al., submitted to Nature Communications.

The manuscript describes the formation of C-H bonds and in particular, the time scale and molecular level control, which they claim remained elusive. The team investigated the light-induced bimolecular reaction using a van der Waals dimer composed of two molecules, H₂ and CO. The pump-probe technique used a two-color photons scheme with a reaction microscope or COLTRIMS to identify the pathways leading to C-H photobonding and producing HCO⁺. The team reports an ultrafast formation time of 198 fs ± 16 fs.

Comment #1. *The paper is well written and the literature is cited sufficiently but the results are not noteworthy.*

Reply: We appreciate the Reviewer for reviewing our manuscript and providing the feedback on our work. Apart from giving the good assessments on the writing and literature citation, the Reviewer mainly concerns the novelty of our results.

As is known, the C-H bond formation from reactions of two small H₂ and CO molecules is the most fundamental inorganic bimolecular reaction occurring in the interstellar clouds that leads to the production of organic matter. Exploring such reaction would offer valuable insights into the transition from inorganic to organic matter and establishes connections between interstellar chemical networks and the origins of life in the universe. However, previous studies on similar reaction associated with C-H bond formation were mostly based on spectroscopy or beam scattering measurements, where the detailed dynamics of C-H bond formation is largely unknown. As we have clarified to Reviewer #2 (please also see our response to Reviewer #2), the most important merit of our work is that we studied this reaction at a molecular level based on the approach of light-induced bimolecular reaction starting from van der Waals dimer, which allows timing and controlling the ultrafast dynamics.

In our work, the timing information for the ultrafast formation dynamics of C-H bond from bimolecular reaction of H₂ and CO was obtained by performing femtosecond pump-probe experiment in a reaction microscope, which is hard to achieve in the previous beam scattering measurement (see refs. J. Phys. Chem. Lett. 7, 2742-2747 (2016), J. Chem. Phys. 77, 5847-5848 (1982)). By using a tailored two-color laser field, we clearly identified the reaction pathway leading to C-H bond making thereby producing HCO⁺ ions. In the previous studies (see for instance ref. J. Chem. Phys. 130, 244302 (2009)), the structural isomer HOC⁺ and HCO⁺ cannot be distinguished in the spectroscopy experiment. Moreover, we also achieved coherent control over the formation dynamics of HCO⁺ ion by manipulating its emission direction by tuning the relative phase of the two-color laser fields. Such control of the C-H bond formation has never been reported in the previous studies based on intermolecular collisions and scattering measurements.

The findings presented in our study are not only of great significance for the field of photochemistry, but also astronomy, to understand the most fundamental light-induced inorganic bimolecular processes in interstellar clouds, which has far-reaching implications, particularly in shedding light on the origins of organic compounds in the universe.

To make the clarification of the above-mentioned point regarding the novelty of our work clear to readers, we have improved the abstract and introduction by highlighting the significance of our results, where the following sentences were added/rewrote in the revised manuscript:

“Our findings shed light on the most fundamental inorganic bimolecular reaction occurring in the interstellar clouds, paving the route towards the real-time visualization and coherent control over the dynamics with unprecedented precision.” (abstract)

“The investigation of C-H bond formation from the reaction of small inorganic substances holds particular significance as it provides profound insights into the fundamental transition from inorganic to organic matter and establishes connections between interstellar chemical networks and the emergence of life^{10,11}. Realizing the coherent control of such C-H bond making process represents a significant advancement in the field of chemical organic synthesis.” (line 34-39 on page 2 in the introduction part)

We also have responded to the Reviewer’s other comments and improved the manuscript by following the Reviewer’s suggestions as detailed in the following.

Comment #2. *A very important figure is missing, the one that shows the production of HCO^+ . It is unclear to this reviewer what Fig. 1b shows. It certainly doesn’t show the photoion-photoion (PIPICO) coincidences; i.e., channels from the two-body CE resulting from H_2 and CO. The authors need to provide the PIPICO map with all the channels that result from the ionization and formation of new fragments. Such a map should unambiguously show channels formed from these ions H^+ , H_2^+ , HCO^+ , CO^+ . For example the authors need to provide the PIPICO for these channels (C^+ , O^+); (H^+ , HCO^+). Fig.3 claims to show the dynamics of the (H^+ , HCO^+); i.e., the KER as a function of time delay but this reviewer will need to see first the PIPICO of the channel in question.*

Reply: We feel sorry for the unclear demonstration and description of the photoion-photoion coincidence (PIPICO) spectrum shown in Fig. 1b in the main text. The displayed PIPICO spectrum presents the measured time-of-flight (TOF) coincidence maps of the two ion fragments from the two-body Coulomb explosion (CE) channels, where the horizontal and vertical axis are the TOF of the first and second ion, respectively. In the PIPICO, if the two ion fragments originate from the two-body breakup of the same molecular entity, the signals appear as the sharp diagonal lines.

In Fig. R7, we show the PIPICO spectrum of the measured raw data. As indicated by the number ①-⑥, several two-body fragmentation channels can be identified. For instance, the two-body CE of H_2 and CO monomers produces the ①(H^+ , H^+), and ②(C^+ , O^+) ion pairs, respectively. The direct CE of $\text{H}_2\cdots\text{H}_2$ and $\text{CO}\cdots\text{CO}$ dimers induced by two-site double ionization produces the ion pairs of ③(H_2^+ , H_2^+), and ④(CO^+ , CO^+), respectively. Moreover, the ion yields arising from the bimolecular reaction of H_2 and CO are identified as the ion pairs ⑤(H_2^+ , CO^+) with a mass-to-charge ratio $m/q = 2$ and 28, and ⑥(H^+ , HCO^+) with $m/q = 1$ and 29, respectively. While the vertical and horizontal sharp lines in the Fig. R7 with $m/q = 1, 2$ and 28 are the ion fragments of H^+ , H_2^+ and CO^+ from the false coincidence fragmentation events.

To suppress the false coincidence signal, the recoil momentum rule of $p_{\text{sum}} = |p_{\text{ion1}} + p_{\text{ion2}}| < 4$ a.u. is imposed for the sum-momentum of coincidentally detected two ion fragments of each channel. The Fig. 1b in the old version of manuscript shows the PIPICO spectrum with the selected real

coincidence ion signals in (H^+ , HCO^+), (H_2^+ , CO^+), (H^+ , H^+), and (C^+ , O^+) channels. By selecting the real coincidence ion signals of (H^+ , HCO^+) channel, the obtained KER distribution of the ion fragments as a function of pump-probe time delay shown in Fig. 3 thus allows one to inspect the ultrafast dynamics of the C-H bond formation in (H^+ , HCO^+) channel.

To make it clear to readers, as suggested by the Reviewer, in the revised new manuscript, we have updated the Fig. 1 by adding the PIPICO spectrum with both the raw data and the selected interested real coincidence ion signals, and rewriting the following sentences.

“Figures 1b presents the measured photoion-photoion coincidence (PIPICO) spectrum for the two-body Coulomb explosion channels, where the horizontal and vertical axes represent the time-of-flight (TOF) of the first and second ion, respectively. The selected signals of interest are plotted in Fig. 1c.” (line 103-106 on page 4)

Moreover, we have added the information of how we extract the ion signals in different fragmentation channels in the Supplemental Materials, which includes a more detailed description for the measured PIPICO spectrum (please also see our response to Reviewer #1).

Fig. R7. Measured PIPICO spectrum with raw data. The several two-body CE channels are indicated by different numbers.

Comment #3. This manuscript is not suitable for publication in Nature Communication because formation of C-H bond is not new and measuring its time dynamics is not sufficient. I would recommend that this paper be submitted instead to J. Chem. Phys. where there is already a body of work regarding this research as cited by the authors. I would also recommend that the authors include the PIPICO map of all the channels in their revised manuscript to be submitted elsewhere as other referees will require it.

Reply: We agree with the Reviewer that the C-H bond formation in producing HCO^+ radical from bimolecular reactions has been studied in the past, for instance, in refs 18-23 in the main text. However, these studies primarily rely on the cross-beam ion-neutral reactions in reactive scattering measurements or theoretical simulations. The extracted information in these steady-state studies is mostly the reaction cross sections, the kinetic energies and the angle-resolved state distribution

of the product containing C-H bonds.

As we have replied to the Reviewer's first comment (please also see our response to Reviewer #2), the novelty of our work is that we performed time-resolved study on the ultrafast dynamics of the C-H bond formation via a light-induced bimolecular reaction starting from a H₂-CO dimer. Since the spatiotemporal starting point of the light-driven bimolecular in a dimer system is well-defined, the ultrafast dynamics of the bimolecular reaction can be precisely tracked in a pump-probe experiment.

In addition to the timing of the C-H bond formation dynamics, we achieved the coherent control over the bimolecular reaction dynamics by manipulating the emission direction of relevant HCO⁺ ion using a tailored two-color laser field. The using of phase-controlled two-color laser fields also allows us to identify the reaction channels responsible for C-H bond formation. This advances the previous scattering experiment where the formed structural isomer HOC⁺ and HCO⁺ is hard to be distinguished (see for instance ref. J. Chem. Phys. 130, 244302 (2009)).

Moreover, the reaction of the two small H₂ and CO molecule in forming C-H bond is the most fundamental inorganic bimolecular reaction occurring in the interstellar clouds that leads to the production of organic matter. Exploring such reaction can offer valuable insights into the transition from inorganic to organic matter and establishes connections between interstellar chemical networks and origins of life. Our findings not only provide insights into the timing of such reaction, but also pave the way towards shaping the bimolecular reaction dynamics with unprecedented precision using ultrafast laser pulses. Considering the mentioned novelty of our work, we are therefore confident that our work will ignite significant interest in the scientific community and appeal to the broad readership of Nature Communications.

Minor remarks:

Typo, like 33, the word thereinto, doesn't belong in the sentence.

Reply: We thank the Reviewer for noticing us this phrase problem. We have corrected the sentence in the revised manuscript.

To summarize, we appreciate the Reviewer for the valuable comments and suggestions on our manuscript. In this response letter, we have clarified the novelty of our work for the Reviewer and accordingly strengthen the introduction and discussion in the revised manuscript. We have also improved the presentation of the PIPICO map by showing the raw data with all the detected channels. We hope our clarification and the improved manuscript can convince the Reviewer that our resubmitted manuscript meets the criteria for publication in Nature Communications.

REVIEWERS' COMMENTS

Reviewer #1 (Remarks to the Author):

The authors have addressed all the concerns I raised and provided detailed explanations and revisions to the manuscript. Given these revisions, I find the results convincing and believe the manuscript is now suitable for publication in Nature Communications.

Reviewer #2 (Remarks to the Author):

The authors have given a very thorough response to all comments, which confirms my general impression: This is a very solid piece of work of high standard.

In my opinion, from the point of view of fundamental molecular reaction dynamics, the real-time observation of C-H bond formation is not fundamentally different from previous observations of bond formation starting from van der Waals molecular dimers (as, e.g., reported in Refs. 34-36 of the manuscript). The authors write, "Our findings shed light on the most fundamental inorganic bimolecular reaction occurring in the interstellar clouds, ...". In interstellar clouds, chemical reactions are initiated by ultraviolet or cosmic radiation. These conditions are clearly different from the laboratory conditions of this work.

Thus, the authors are still not convincing me that their results are sufficiently groundbreaking to warrant publication in Nature Communications.

Reviewer #3 (Remarks to the Author):

The authors have addressed all my concerns and, in my opinion, those of the other reviewers. They present noteworthy results, showing that H₂ and CO⁺ form a chemical bond within ~200fs. This is important fundamental information of significance to astrochemistry.

Including the PIPICO map in Fig. 1c is very good, however, it should be labeled as in figure R7. More importantly, it should be clear which features belong to the fast and slow channels being discussed in the manuscript.

Author Response to Reviewer's Comments:

We thank all the Reviewers for agreeing to review our manuscript a second time and for their thoughtful comments. We have revised the manuscript according to their suggestions.

Below, we reproduce the original *comments by the Reviewers in italics and black*, provide **our responses in blue** and the **changes to the manuscript in red**.

Reply to Reviewer #1

The authors have addressed all the concerns I raised and provided detailed explanations and revisions to the manuscript. Given these revisions, I find the results convincing and believe the manuscript is now suitable for publication in Nature Communications.

Reply: We very much appreciate Reviewer #1 for his/her time on reviewing our manuscript and for recommending the publication of our work in Nature Communications.

Reply to Reviewer #2

The authors have given a very thorough response to all comments, which confirms my general impression: This is a very solid piece of work of high standard.

Reply: We very much appreciate Reviewer for reviewing our manuscript a second time and for the positive remarks on our revised new manuscript. We have addressed the remaining concerns of the Reviewer #2 in the following.

In my opinion, from the point of view of fundamental molecular reaction dynamics, the real-time observation of C-H bond formation is not fundamentally different from previous observations of bond formation starting from van der Waals molecular dimers (as, e.g., reported in Refs. 34-36 of the manuscript). The authors write, "Our findings shed light on the most fundamental inorganic bimolecular reaction occurring in the interstellar clouds, ...". In interstellar clouds, chemical reactions are initiated by ultraviolet or cosmic radiation. These conditions are clearly different from the laboratory conditions of this work. Thus, the authors are still not convincing me that their results are sufficiently groundbreaking to warrant publication in Nature Communications.

Reply: As we have explained to the Reviewer #2 in the first-round response, the novelty of our work is that we performed the molecular-level study of the C-H bond formation dynamics in a bimolecular reaction from two small inorganic molecules. Exploring the formation of C-H bonds through reactions involving small inorganic molecules would offer valuable insights into the fundamental transition from inorganic to organic matter and thus the origins of life. In our work, we focus on the C-H bond formation from the reaction of two small H₂ and CO molecules. As compared to the previous beam scattering studies, the C-H bond formation in this work is initiated by light-induced bimolecular reaction starting from a van der Waals (vdW) dimer H₂-CO formed at

a very low temperature condition. The approach of light-induced bimolecular reaction from vdW dimer ensures a well-defined spatiotemporal starting point for time-resolved studies. The using of waveform-controlled ultrashort laser light as irradiation source not only can provide femtosecond time resolution for the time-resolved studies, but also allows achieving the coherent control of the C-H bond formation dynamics.

We agree with the Reviewer that the irradiation source for the bimolecular reaction in this work, is not completely the same as that in the interstellar media, where the molecular reactions could be mostly initiated by the ultraviolet light radiation or cosmic radiation. Considering this, in the revised new manuscript, we have weakened the relation of our current study with the astrochemical reaction in the interstellar clouds. Since the involved H₂ and CO molecules represent two prevalent small inorganic substances in the universe, the real-time visualization, and the coherent control over of the reaction dynamics in forming C-H bond under our laboratory conditions still hold significant promise for understanding the most fundamental inorganic bimolecular reactions responsible for the fundamental transition from inorganic to organic matter. Therefore, we believed that our work meets the criteria of publication in Nature Communications.

To make it clear and avoid any confusion to the readers, we have rephrased the following sentences in the revised manuscript:

“The real-time visualization and coherent control of the dynamics contribute to a deeper understanding of the most fundamental bimolecular reactions responsible for C-H bond formation, thus elucidating the emergence of organic components in the universe.” (abstract)

“Our molecular-level study of the C-H bond formation at low temperatures casts light on the underlying light-induced inorganic bimolecular reaction processes, and carries substantial potential for comprehending the pivotal transition from inorganic to organic matter, thereby providing insights into the origins of the abundant organic compounds distributed throughout the universe.” (line 296-303 on page 11)

Reply to Reviewer #3

The authors have addressed all my concerns and, in my opinion, those of the other reviewers. They present noteworthy results, showing that H₂ and CO⁺ form a chemical bond within ~200fs. This is important fundamental information of significance to astrochemistry.

Reply: We very much appreciate Reviewer for these very positive remarks on our revised new manuscript and for the recommendation. We have addressed the remaining concerns of the Reviewer in the following.

Including the PIPICO map in Fig. 1b is very good, however, it should be labeled as in figure R7. More importantly, it should be clear which features belong to the fast and slow channels being discussed in the manuscript.

Reply: We thank the Reviewer for this comment. We have updated the PIPICO map in Fig. 1b, where all the involved two-body fragmentation channels are labeled, as that shown in Fig. R7 in

the first-round response letter. Regarding the second concern, as mentioned by the Reviewer, there are fast and slow pathways in (H^+ , HCO^+) channels, which indeed can be clearly distinguished in the raw data because the ion fragments with high and low kinetic energy release (KER) during the fragmentation will be registered on the detector with different time-of-flight (TOF) or impact position X/Y information. Since the light-induced Coulomb explosion process mostly occur along the laser polarization direction, the signal of ejected ion fragments with same mass but with different KER are mostly distinguishable along the laser polarization direction. If the laser polarization is along z direction, i.e., the TOF direction, the slow and fast pathway of (H^+ , HCO^+) channels can be clearly distinguished in the PIPICO spectrum, as featured at the central part (slow pathway) and shoulder part (fast pathway) of the sharp diagonal line. The shown PIPICO spectrum in Fig. 1 is measured using laser pulse with polarization along y direction. In such case, as shown in Fig. R1, the signals of ion fragments with same mass from slow and fast pathway are overlapped in TOF axis, which thus cannot be distinguished in the shown PIPICO spectrum. However, since the ion fragments from the two pathways have different emission momentum along y direction, the slow and fast pathway of (H^+ , HCO^+) channels can be easily distinguished when considering both the position Y and TOF information.

Fig. R1 Measured position Y-TOF spectrum of the ion fragments produced in (H^+ , HCO^+) channel. The slow (inner ring) and fast (outer ring) pathway are indicated by arrows.

To make it clear to readers, we have added Fig. R1 to the revised Supplementary Information, and added the following sentences:

“For the investigated (H^+ , HCO^+) channel, the fast and slow pathways can be clearly distinguished in the raw data because the ion fragments with high and low KER during the fragmentation will be registered on the detector with different time-of-flight (TOF) or impact position X/Y information. If the laser polarization is along z direction, i.e., the TOF direction, the slow and fast pathway can be clearly distinguished in the PIPICO spectrum, as featured at the central part (slow pathway) and shoulder part (fast pathway) of the sharp TOF correlation diagonal line. While when the laser polarization is along y direction, as shown in Fig. 1c of the main text, the signals of ion fragments with same mass from slow and fast pathway are overlapped in the TOF axis, which thus cannot be distinguished in the PIPICO spectrum. In such case, however, since the ion fragments from the two pathways have different emission

momentum along y direction, as shown in Supplementary Fig. 1, the slow and fast pathway of (H^+ , HCO^+) channels can be easily distinguished when considering both the position Y and TOF information.” (line 79-91 on page5)